# Sniff-synchronized, gradient-guided olfactory search by freely moving mice

Teresa M Findley[1†], David G Wyrick[1†], Jennifer L Cramer[2], Morgan A Brown[2], Blake Holcomb[2], Robin Attey[2], Dorian Yeh[2], Eric Monasevitch[2], Nelly Nouboussi[2], Isabelle Cullen[2], Jeremea O Songco[1], Jared F King[2], Yashar Ahmadian[1,3]*, Matthew C Smear[2]*

[1]Department of Biology and Institute of Neuroscience, University of Oregon, Eugene, United States; [2]Department of Psychology and Institute of Neuroscience, University of Oregon, Eugene, United States; [3]Computational & Biological Learning Lab, University of Cambridge, Cambridge, United Kingdom

**Abstract** For many organisms, searching for relevant targets such as food or mates entails active, strategic sampling of the environment. Finding odorous targets may be the most ancient search problem that motile organisms evolved to solve. While chemosensory navigation has been well characterized in microorganisms and invertebrates, spatial olfaction in vertebrates is poorly understood. We have established an olfactory search assay in which freely moving mice navigate noisy concentration gradients of airborne odor. Mice solve this task using concentration gradient cues and do not require stereo olfaction for performance. During task performance, respiration and nose movement are synchronized with tens of milliseconds precision. This synchrony is present during trials and largely absent during inter-trial intervals, suggesting that sniff-synchronized nose movement is a strategic behavioral state rather than simply a constant accompaniment to fast breathing. To reveal the spatiotemporal structure of these active sensing movements, we used machine learning methods to parse motion trajectories into elementary movement motifs. Motifs fall into two clusters, which correspond to investigation and approach states. Investigation motifs lock precisely to sniffing, such that the individual motifs preferentially occur at specific phases of the sniff cycle. The allocentric structure of investigation and approach indicates an advantage to sampling both sides of the sharpest part of the odor gradient, consistent with a serial-sniff strategy for gradient sensing. This work clarifies sensorimotor strategies for mouse olfactory search and guides ongoing work into the underlying neural mechanisms.

*For correspondence:
ya311@cam.ac.uk (YA);
smear@uoregon.edu (MCS)

†These authors also contributed equally to this work

Competing interests: The authors declare that no competing interests exist.

## Introduction

Sensory observations are often made in concert with movements (*Ahissar and Assa, 2016*; *Gibson, 1966*). During active search behavior, animals make sampling movements in order to extract relevant sensory information from the environment (*Gibson, 1962*; *Schroeder et al., 2010*). Sampling behavior is flexible and can be customized for the problem the animal is trying to solve (*Kleinfeld et al., 2006*; *Yarbus, 1967*). In the brain, sensory and motor systems interact extensively (*Andersen and Mountcastle, 1983*; *Duhamel et al., 1992*; *McGinley et al., 2015*; *Musall et al., 2019*; *Niell and Stryker, 2010*; *Poulet and Hedwig, 2006*; *Sommer and Wurtz, 2002*; *Stringer et al., 2019*), which reflects the importance of interpreting self-induced stimulus dynamics (*Sommer and Wurtz, 2008*; *Sperry, 1950*; *von Holst and Mittelstaedt, 1950*; *Webb, 2004*). Here, we show how mice sample the environment while navigating a noisy odor gradient.

Navigating by chemical cues may be one of the most ancient problems motile organisms evolved to solve, and it remains crucial in the lives of almost all modern species. Unicellular organisms and some invertebrates navigate chemical gradients by chemotaxis (*Bargmann, 2006*; *Berg, 2000*;

*Lockery, 2011*). In essence, their movement programs can be described as having two states: they move straight when the concentration is increasing and reorient their movements when the concentration is decreasing. Whereas chemical gradients are stable and informative at the spatial scale of these organisms, for many larger or flying organisms, odor gradient cues do not provide useful positional information (*Crimaldi et al., 2002*; *Murlis et al., 1992*; *Baker et al., 2018*). At this larger spatial scale, turbulent airflow moves odor molecules in dynamic spatiotemporal patterns, disrupting concentration gradients and nullifying classical chemotaxis strategies. Instead, olfactory cues often gate movements that depend on other sensory modalities. Here too, these organisms' behavioral structure can be described as transitions between two states: detection of odor promotes upwind movement while the absence of odor promotes crosswind casting movement (*Kennedy and Marsh, 1974*; *van Breugel and Dickinson, 2014*; *Vickers and Baker, 1994*). In this behavioral program, known as odor-gated anemotaxis, odor cues gate behavioral responses to positional information provided by another modality. In both chemotaxis and odor-gated anemotaxis, search tasks can be described with a two-state search model.

In comparison to invertebrates, our understanding of olfactory search behavior in vertebrates is more rudimentary, even in commonly studied rodent models. In these animals, access to the olfactory environment is gated by respiration, which is in turn responsive to incoming olfactory stimulation (*Kepecs et al., 2006*; *Wachowiak, 2011*). Novel odors evoke rapid sniffing, during which respiration synchronizes with whisker, nose, and head movements on a cycle-by-cycle basis (*Kurnikova et al., 2017*; *Moore et al., 2013*; *Ranade et al., 2013*). Thus, during active mammalian olfaction, sensory and motor systems interact in a closed loop via the environment, as is true for other sensory modalities such as vision or somatosensation (*Ahissar and Assa, 2016*; *Gibson, 1966*). The cyclical sampling movements coordinated by respiration further synchronize with activity in widespread brain regions (*Karalis and Sirota, 2018*; *Kay, 2005*; *Macrides et al., 1982*; *Vanderwolf, 1992*; *Yanovsky et al., 2014*; *Zelano et al., 2016*) similarly to correlates of locomotor, pupillary, and facial movements observed throughout the brain (*McGinley et al., 2015*; *Musall et al., 2019*; *Niell and Stryker, 2010*; *Stringer et al., 2019*). Respiratory central pattern generators may coordinate sampling movements to synchronize sensory dynamics across modalities with internal brain rhythms (*Kleinfeld et al., 2014*).

Previous work has shown that rodents follow odor trails, where the concentration gradient is steep and stable, with rapid sniffing accompanied by side-to-side head movements (*Jones and Urban, 2018*; *Khan et al., 2012*). In these conditions, serial sniffing and stereo olfactory cues guide movements of the nose. Likewise, moles used concentration comparisons across space and time to locate a food source in a sealed experimental chamber in which a lack of airflow allowed for even diffusion of a chemical gradient (*Catania, 2013*). In this study, when input to the nares was reversed, moles navigated towards odor sources at a distance, but demonstrated significant deficits at identifying odor location when near the source. Behavioral modeling in mice further supports that internaris concentration comparison plays a more important role in search near the source (*Liu et al., 2020*). Thus, both serial sniffing and stereo cues can guide olfactory search behavior. The sensory computations and movement strategies employed during navigation of an airborne odor plume are less clear. In previous experiments where rodents searched in airborne odor plumes, mice developed a memory-based strategy of serially sampling each possible reward location for the presence of odor, turning search tasks into detection tasks (*Bhattacharyya and Bhalla, 2015*; *Gire et al., 2016*). Thus, it remains unclear whether mammals can follow noisy concentration gradients under turbulent conditions.

To better understand the sensory computations and sampling strategies for olfactory search, we designed a two-choice behavioral assay where mice use olfactory cues to locate an odor source while we monitor sniffing and movements of the head, nose, and body. We found that mice use a concentration gradient-guided search strategy to navigate olfactory environments that contain turbulent flow. We found that these navigational behaviors are robust to perturbations including introduction of a novel odorant, varying the concentration gradient, and naris occlusion. Given the fundamental importance of sniffing to olfactory function, we hypothesized that mice would selectively sample the environment such that nose movement would be tightly coupled to respiration. Consistent with this hypothesis, we found that mice synchronize rhythmic three-dimensional head movements with the sniff cycle during search. These sniff-synchronized movement rhythms are prominent during trials, and largely absent during the inter-trial interval (ITI), suggesting that sniff

synchronous movement is a proactive strategy rather than a reactive reflex. To find structure in this search strategy, we used unsupervised computational methods to parse movement trajectories into discrete motifs. These movement motifs are organized into two distinguishable behavioral states corresponding to investigation and approach, reminiscent of the two-state olfactory search programs described in smaller organisms. Temporally, investigation motifs lock to the sniff cycle with precision at a tens of milliseconds scale. Spatially, patterns of investigation and approach usage indicate a strategic advantage for investigating across the steepest part of the odor gradient. Our findings reveal the microstructure of olfactory search behavior in mice, identifying sensory computations and movement strategies that are shared across a broad range of species.

## Results

### Olfactory search in noisy gradients of airborne odor

We developed a two-alternative choice task in which freely moving mice report odor source location for water rewards (Materials and methods and *Figure 1A*). To capture the search behavior, we measured respiration using nasal thermistors (*McAfee et al., 2016*) and video-tracked the animal's body, head, and nose position in real time at 80 frames/s (*Figure 1B, E* and *Figure 1—figure supplement 1*). The mouse initiates a trial by inserting its nose in a port (*Figure 1C*, 'initiation'), which activates odor release from two ports at the opposite end of the arena. The mouse reports the location of higher odor concentration by walking toward it (*Figure 1C*, 'search'). In previous studies, rodents performing olfactory search tasks developed memory-guided foraging strategies. In essence, animals run directly to potential odor sources and sample each in turn, thus converting the search tasks to detection tasks (*Bhattacharyya and Bhalla, 2015*; *Gire et al., 2016*). To prevent mice from adopting sample-and-detect strategies, our task forces mice to commit to a decision at a distance from the actual source. Using real-time video-tracking (*Lopes et al., 2015*), we enforced a virtual 'decision line', such that the trial outcome is determined by the mouse's location when it crosses this decision line (*Figure 1C*, 'outcome'). For stimuli, we deliver odor from two separate flow-dilution olfactometers, giving independent control over odor concentration on the two sides. To test olfactory search over a range of difficulties, we presented four odor patterns, defined by the ratio of odor concentration released from the two sides (*Video 1*, 100:0,80:20, 60:40, 0:0).

We measured the spatiotemporal distribution of odor using a photoionization detector (PID) in a $5 \times 7$ grid of sampling locations (*Figure 1D* and *Figure 1—figure supplement 2*). Pinene was used for the majority of experiments because it is a neutral-valence odorant that is sensitively detected by the PID. As designed, varying the concentration ratios produced across-trial-averaged gradients of different magnitudes. Airflow in the arena is turbulent, imposing temporal fluctuations on the odor gradient (*Video 2*). Thus, our assay tests an animal's ability to navigate noisy odor gradients.

Mice learn the olfactory search task rapidly and robustly. We trained mice in the following sequence (*Figure 2A*): first, naïve, water-restricted mice obtained water rewards from all ports in an alternating sequence (*Figure 2—figure supplement 1A*; 'Water sampling'). In the next phase of training, we added odor stimulation such that odor delivery alternated in the same sequence as reward, so that the mice would learn to associate odor with reward ports (*Figure 2—figure supplement 1B*; 'Odor association'). Following these initial training steps, mice were introduced to the olfactory search paradigm. Odor was pseudo-randomly released from either the left or right odor source ('100:0'), signaling water availability at the corresponding reward port. Almost all mice performed above chance in the first session (*Figure 2B*; binomial test, p<0.05 for 24 out of 25 mice, 75 ± 9.2% correct, mean ± sd). Within four sessions, most animals exceeded 80% performance (19 out of 26). Following 100:0, mice were introduced to the 80:20 condition with mean performance across mice in the first session reaching ~60% (*Figure 2—figure supplement 1*). Most subjects improved to exceed 70% performance over the next seven sessions (17 out of 24). The mice that did not were excluded from subsequent experiments.

Next, we tested whether mice trained to search pinene plumes would generalize their search behavior to a novel odorant. We chose vanillin as the novel odorant because, unlike pinene, vanillin does not activate the trigeminal fibers of the nose (*Cometto-Muñiz and Abraham, 2010*; *Doty et al., 1978*; *Hummel et al., 2009*). Thus, we could test whether trigeminal chemosensation is necessary for performance in our task. We found no differences in performance between vanillin and

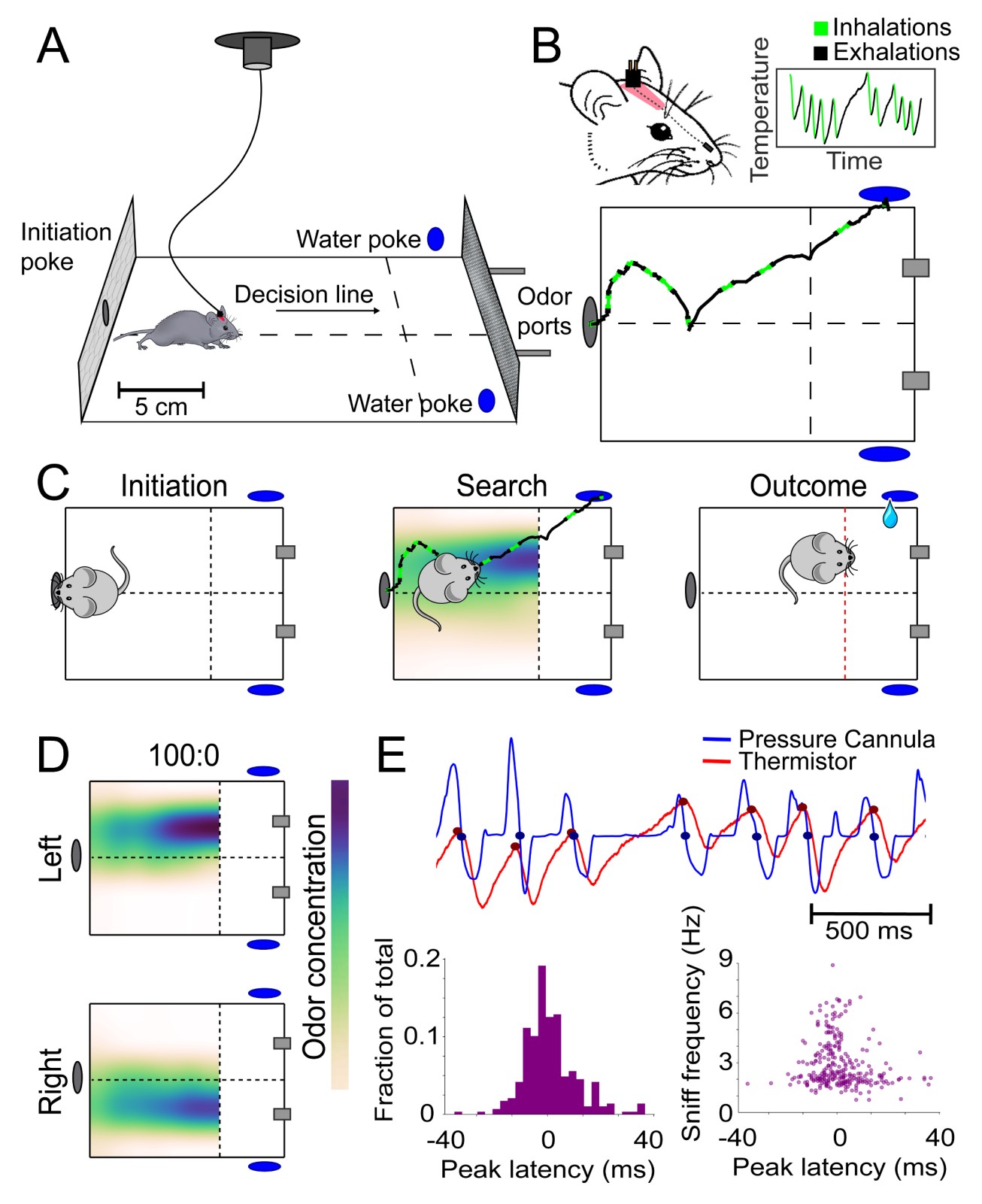

**Figure 1.** Behavioral assay for freely moving olfactory search. (**A**) Diagram of experimental chamber where mice are tracked by an overhead camera while performing olfactory search. (**B**) Top: nose and head positions are tracked using red paint at the top of the head. Sniffing is monitored via an intranasally implanted thermistor. Bottom: example of sniffing overlaid on a trace of nose position across a single trial. (**C**) Diagram of trial structure. Initiation. Mice initiate a trial via an initiation poke (gray oval). Search. Odor is then released from both odor ports (gray rectangles) at different

*Figure 1 continued on next page*

*Figure 1 continued*

concentrations. Outcome. Mice that cross the decision line (red) on the side delivering the higher concentration as tracked by the overhead camera receive a reward at the corresponding water port (blue ovals). (D) Colormaps of average odor concentration across ~15 two-second trials captured by a 7 × 5 grid of sequential photoionization detector recordings. Rows represent side of stimulus presentation (left or right). Odor concentrations beyond the decision line were not measured. (E) Comparison of sniff recordings taken with an intranasally implanted thermistor and intranasally implanted pressure cannula. These are implanted on the same mouse in different nostrils. Top: example trace of simultaneous pressure cannula (blue) and thermistor (red) recordings with inhalation points (as detected in all future analyses) overlaid on the traces in their respective colors. Bottom left: histogram of peak latencies (pressure inhalation onset – thermistor inhalation onset). 14/301 inhalations (4.7%) were excluded as incorrect sniff detections. These were determined as incorrect because they fell more than 2 standard deviations outside the mean in peak latency (mean = 1.61585 ms, SD = ±14.93223 ms). Bottom right: peak latencies, defined as the difference between pressure inhalation onset and thermistor inhalation onset, plotted against instantaneous sniff frequency.

The online version of this article includes the following figure supplement(s) for figure 1:

**Figure supplement 1.** Calibrating alignment of video frames with sniff signal.
**Figure supplement 2.** Characterizing the odor stimulus conditions.

pinene sessions for these mice (*Figure 2—figure supplement 2A*; Wilcoxon rank-sum test, p=0.827, *n* = 3). These data suggest that this search behavior generalizes across odors and does not rely on the trigeminal system.

## Mice can use gradient cues in turbulent flow

We reasoned that mice would solve this task using odor gradient cues. To vary odor gradients between trials, we trained mice in sessions with interleaved concentration ratios (100:0, 80:20, 60:40) across the trials of a session. In addition to these concentration ratios, odor omission probe trials (0:0) were randomly interleaved into all experimental sessions. During these trials, airflow was identical to 80:20 trials, but air was directed through an empty vial rather than a vial containing odorant solution. These odor omission trials served a twofold purpose: they acted as controls to ensure behavior was indeed odor-guided, and they allowed us to observe how absence of odor impacts search behavior. On these probe trials, mice performed at chance (binomial test, p=0.9989), with longer trial durations (Wilcoxon rank-sum test, p<0.05) and more tortuous trajectories (Wilcoxon rank-sum test, p<0.05) than on non-probe trials (*Figure 2C*; *n* = 19, all data from 80:20 condition with probe trials). Performance drops with the concentration ratio (ΔC), consistent with our reasoning that mice would use odor gradient cues in this task (*Figure 2D*; pairwise Wilcoxon rank-sum tests, p<0.05, *n* = 15). Varying the concentration ratio from 80:20 to 60:40 did not affect trial duration or path tortuosity, defined as actual path length divided by direct path length (*Figure 2D*; pairwise Wilcoxon rank-sum tests, p>0.05). However, trial duration and path tortuosity were slightly, but statistically significantly, longer in the 100:0 condition (pairwise Wilcoxon rank-sum tests, p<0.05).

Given that these results were obtained using a single absolute concentration (|C|) across ratios, mice could be solving our task with two distinct categories of sensory computation. One possibility is that information about source location is extracted from the odor gradient. An alternative strategy would be to make an odor intensity judgment that gates a response to positional information from non-olfactory cues, such as wind direction, visual landmarks, or self-motion. This computation would be reminiscent of the odor-gated visual and mechanosensory behaviors observed in insects (*Álvarez-Salvado et al., 2018*; *Kennedy and Marsh, 1974*; *van Breugel and Dickinson, 2014*). To distinguish between these possible strategies, we tested mice in sessions interleaving the air dilution ratios 90:30 and 30:10. 30 is the correct answer in one condition and incorrect in the other, so that mice cannot use an intensity judgment strategy to perform well in both ratio conditions. In both conditions,

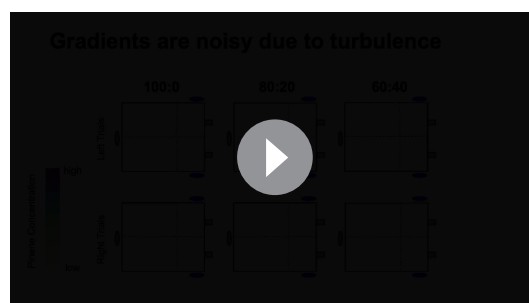

**Video 1.** Odor gradients are temporally dynamic and noisy. Colormaps represent the time course of odor concentration for pseudo-trials assembled from individual trials at each sampling location.
https://elifesciences.org/articles/58523#video1

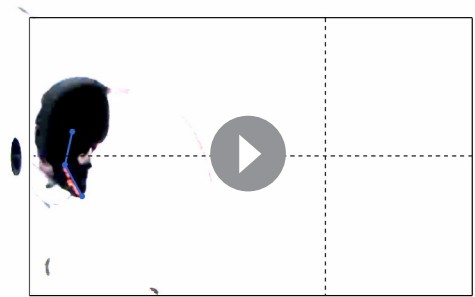

**Video 2.** Example trials with sniffing. Three dots on the mouse represent the coordinates of front of snout, back of head, and center of mass extracted using Deeplabcut. Sniffing is indicated by color (blue = inhalation, pink = exhalation) and sound (higher tone = inhalation, lower tone = exhalation). Video frame rate is slowed by 4×.
https://elifesciences.org/articles/58523#video2

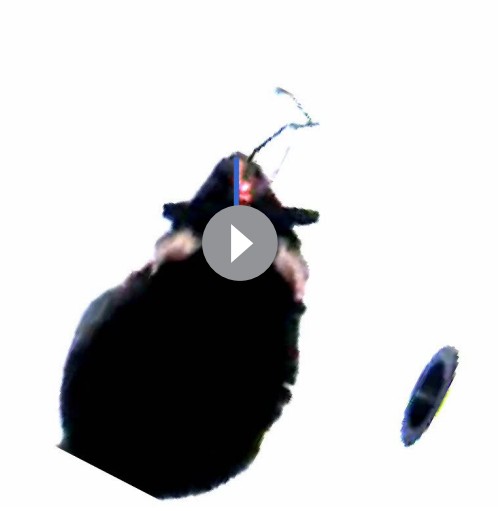

**Video 3.** Movement trajectories for individual sniffs. Each video snippet corresponds to one sniff, where the frames are translated so that the back of the head is centered, and rotated so that the head angle is vertical, in the first frame of each sniff. Blue = inhalation, pink = exhalation. Video frame rate is slowed by 10×.
https://elifesciences.org/articles/58523#video3

mice performed equally well in the first session of training (*Figure 2E*; Wilcoxon rank-sum test, p=0.465, *n* = 5). This equal performance is true within the first 20 trials of the session (*Figure 2—figure supplement 2*; Wilcoxon rank-sum test, p=0.296). These results indicate that odor gradients guide olfactory search under these conditions.

We next asked how the mice are sensing the concentration gradient. Many mammals can use stereo olfaction: comparing odor concentration samples between the nares (*Catania, 2013*; *Parthasarathy and Bhalla, 2013*; *Porter et al., 2007*; *Rabell et al., 2017*; *Rajan et al., 2006*). To test the role of stereo comparisons in our olfactory search task, we performed naris occlusion experiments. Mice were tested in three conditions on alternating days: naris occlusion, sham occlusion, and no procedure. We found that naris occlusion did not significantly impact performance or path tortuosity (pairwise Wilcoxon rank-sum tests, p>0.05). When compared with the no-stitch condition, the naris stitch condition resulted in a slight, but statistically significant, increase in trial duration (pairwise Wilcoxon rank-sum test, p<0.05).

This is not true when the stitch condition is compared with the sham condition (pairwise Wilcoxon rank-sum test, p>0.05), indicating this may be a result of undergoing a surgical procedure. These overall results indicate that stereo comparison is not necessary in this task (*Figure 2F*; *n* = 13), and that temporal comparisons across sniffs (*Catania, 2013*; *Parabucki et al., 2019*) play a larger role under our task conditions.

## Sniff rate and occupancy are consistent across trials and gradient conditions

To investigate active sampling over the time course of trials, we tracked the animals' sniffing, position, and posture during behavioral sessions. The overall sniff pattern was consistent across

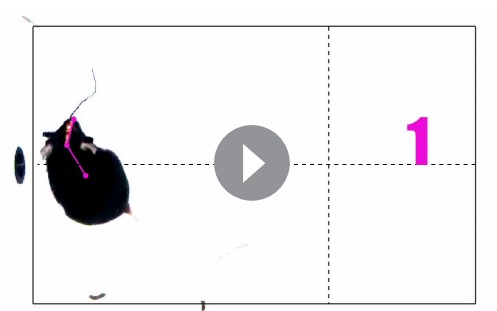

**Video 4.** Example trials with motif sequences. Three dots on the mouse represent the coordinates of front of snout, back of head, and center of mass extracted using Deeplabcut. Dots and lines are colored according to the motif to which that frame was assigned by the auto-regressive hidden Markov model. Video frame rate is slowed by 8×.
https://elifesciences.org/articles/58523#video4

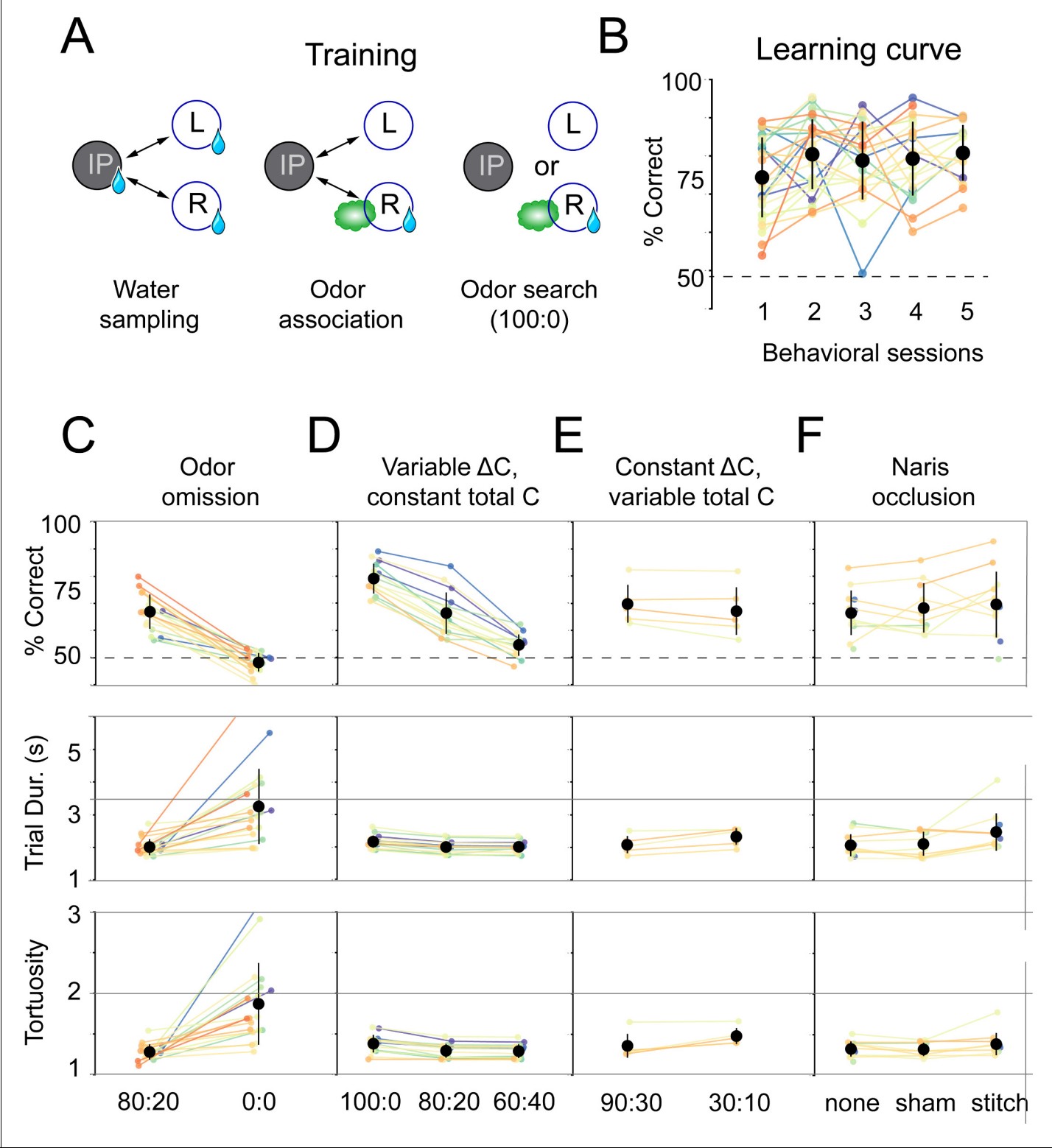

**Figure 2.** Mice use concentration gradient cues in turbulent flow to perform search. (**A**) Initial training steps. Water sampling. In this task, mice alternate in sequence between the initiation, left, and right nose pokes to receive water rewards. Odor association. Next, mice run the alternation sequence as above without water rewards released from the initiation poke, making its only utility to initiate a trial. Further, odor is released on the same side of water availability to create an association between odor and reward. Odor search. Here, mice initiate trials by poking the initiation poke. Odor is then randomly released from the left or right odor port. Correct localization (see *Figure 1C*, decision line) results in a water reward and incorrect is deterred

*Figure 2 continued on next page*

*Figure 2 continued*

by an increased inter-trial interval (ITI). (B) Performance curve across sessions for the odor search (100:0) training step (*n* = 26). (C–F) Session statistics for four different experiments. Each colored line is the average of an individual mouse across all sessions, black points are means across mice, and whiskers are ±1 standard deviation across mice. Top: percent of correct trials. Middle: average trial duration. Bottom: average path tortuosity (total path length of nose trajectory/shortest possible path length). (C) Odor omission. The 80:20 concentration ratio (*Figure 1*) and odor omission (0:0) conditions randomly interleaved across a session. Data shown includes all sessions for each mouse (*n* = 19). (D) Variable Δ*C*, Constant |*C*|. Three concentration ratio conditions (100:0, 80:20, 60:40) randomly interleaved across a session. Data shown includes all sessions for each mouse (*n* = 15). (E) Constant Δ*C*, Variable |*C*|. Concentration ratio conditions 90:30 and 30:10 randomly interleaved across a session (*n* = 5). Data shown for first session only. (F) Naris occlusion. 80:20 sessions for mice with no naris stitch, a sham stitch that did not occlude the nostril, and a naris stitch that occluded one nostril (*n* = 13). Data shown includes all naris occlusion sessions even if the mouse did not perform under every experimental condition.

The online version of this article includes the following figure supplement(s) for figure 2:

**Figure supplement 1.** Session statistics across trainer sessions.

**Figure supplement 2.** Mice generalize search task to novel odorants and variable |*C*| session.

trials, with an inhalation just before trial initiation followed by a long exhalation or pause at the beginning of the trial (*Figure 3*). Next, the mice performed a rapid burst of sniffs, then sniffed more slowly as they approached the target (*Figure 3A*). In this active behavioral state, inhalation and sniff durations were shorter during trials than during ITIs (p<<0.01 for all mice; Kolmogorov–Smirnov test; *Figure 3B, C*), and strikingly shorter than those observed in head-fixed rodents (*Bolding and Franks, 2017*; *Shusterman et al., 2011*; *Wesson et al., 2009*). After the decision, there is a second rapid burst of sniffing followed by a long exhalation or pause during reward anticipation and retrieval (*Figure 3A*). The overall sniff pattern was consistent across trials with an inhalation just before trial initiation followed by a long exhalation or pause at the beginning of the trial (*Figure 3A*). During this sniffing behavior, the mice moved their nose through tortuous trajectories that were not stereotyped from trial to trial (*Figure 3D, E*; *Video 1*). Although individual mice showed position biases (*Figure 3—figure supplement 1*), these biases were not systematic across mice, so that the across-mouse mean occupancy distribution was evenly distributed across the two sides of the arena (*Figure 3F*; *n* = 19). Consistent with this sniffing and movement pattern, the sniff rate was highest near the initiation port and slower on the approach to target (*Figure 3G*). These measures of active sampling were not statistically distinguishable across gradient or naris occlusion conditions, but changed significantly on odor omission probe trials, with more fast sniffing and head turns overall.

## Mice synchronize three-dimensional kinematic rhythms with sniffing during olfactory search

To test the hypothesis that nose movement locks to respiration during olfactory search, we aligned movement dynamics with the sniff signal. Using Deeplabcut (*Mathis et al., 2018*; *Mathis and Mathis, 2020*), we tracked the position of three points: tip of snout, back of head, and center of mass (*Figure 4A*; *Video 3*). From the dynamics of these three points, we extracted the kinematic parameters nose speed, head yaw velocity, and Z-velocity (*Figure 4B–D*). Synchrony between movement oscillations and sniffing is apparent on a sniff-by-sniff basis (*Figure 5*), Video consistent across mice, and selectively executed during olfactory search. On average, nose speed accelerates during exhalation, peaks at inhalation onset, and decelerates during inhalation (*Figure 5Ai*).

Head yaw velocity, which we define as toward or away (*Figure 4*; centripetal or centrifugal) from the body-head axis, reaches peak centrifugal velocity at inhalation, decelerates and moves centripetally over the course of inhalation (*Figure 5Aii*). Although our videos are in two dimensions, we can approximate movement in depth by analyzing the distance between the tip of the snout and the back of the head (*Figure 4B*). This measure confounds pitch angular motion and vertical translational motion, so we conservatively refer to this parameter as 'Z-velocity'. Because mice point their head downward during task performance, shortening of the distance between the tip of the snout and the back of the head indicates downward movement, while increases in the distance correspond to upward movements. The Z-velocity reaches peak upward velocity at inhalation onset, decelerates and goes downward during inhalation, and rises again at exhalation (*Figure 5Aiii*). These modulations were absent from trial-shuffled data (*Figure 5—figure supplement 1*; permutation test, p<0.001). Cross-correlation and spectral coherence analysis further demonstrates the synchrony

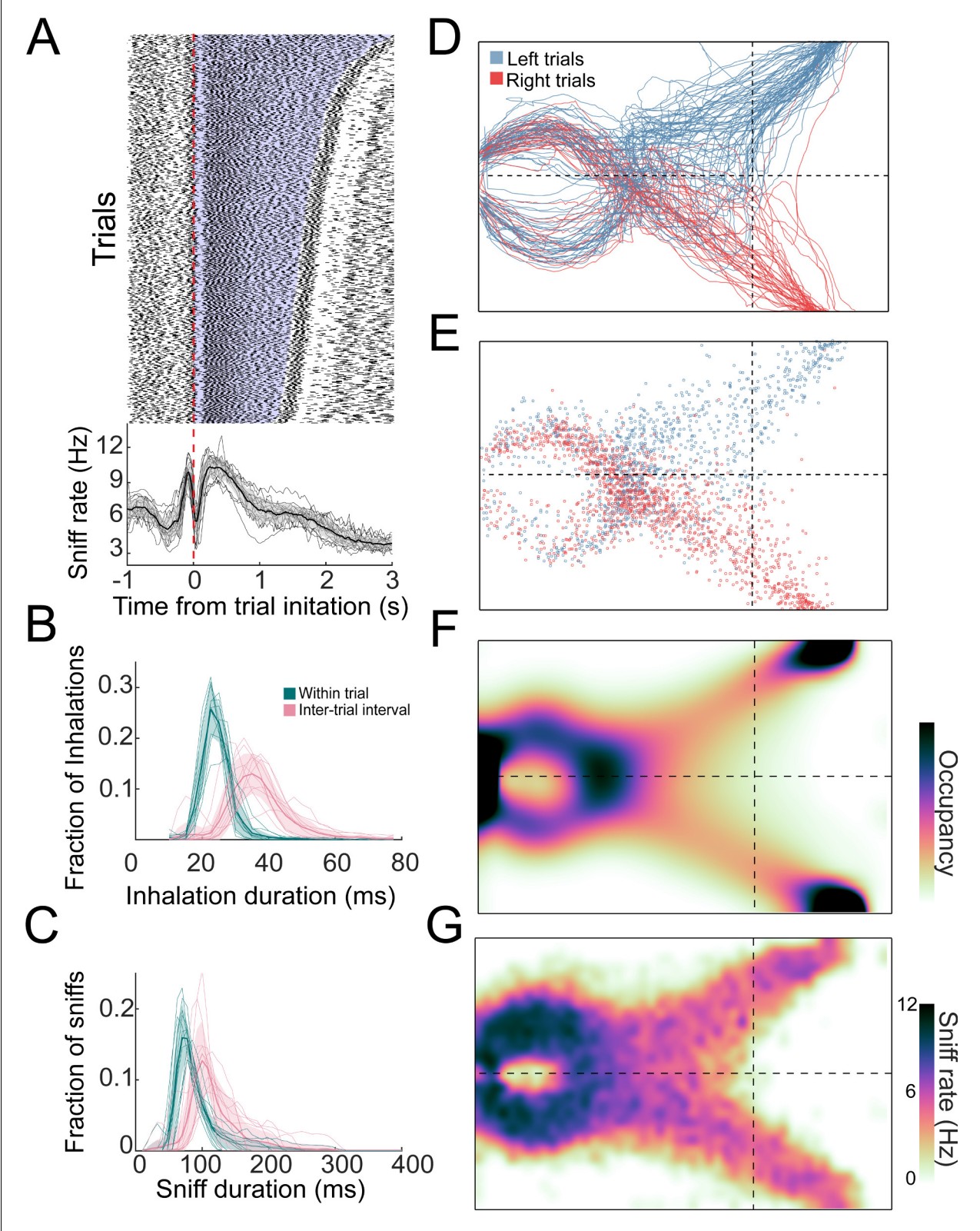

**Figure 3.** Distributions of sniffs and nose positions during search task. (**A**) Above: sniff raster plot for three sessions. Each black point is an inhalation, each row is a trial aligned to trial initiation (dashed line). Rows are sorted by trial length. Blue region represents trial initiation to trial end. Below: mean instantaneous sniff rate across all trials for all mice aligned to time from trial initiation. Thin lines are individual mice, the thick line is the mean across mice, and shaded region is ±1 standard deviation. (**B**) Histogram of inhalation duration time across all mice (*n* = 11). Thick lines and shaded regions are

*Figure 3 continued on next page*

*Figure 3 continued*
mean and ±1 standard deviation, thin lines are individual mice. Green: within-trial sniffs; pink: inter-trial interval sniffs. (C) Histogram of sniff duration time across all mice (*n* = 11). (D) The nose traces of each trial across a single session, colored by chosen side. (E) Location of all inhalations across a single session, colored by chosen side. (F) Two-dimensional histogram of occupancy (fraction of frames spent in each 0.5 cm² bin). Colormap represents grand mean across mice (*n* = 19). (G) Grand mean sniff rate colormap across mice (*n* = 11).
The online version of this article includes the following figure supplement(s) for figure 3:

**Figure supplement 1.** Idiosyncratic occupancy distributions across individual mice.

between nose movement and sniffing (*Figure 5B, C*). These results demonstrate that kinematic rhythms lock to sniffing with tens of milliseconds precision, consistent with a previous report demonstrating that rats make similar movements during novel odor-evoked investigative behavior (*Kurnikova et al., 2017*). Our findings show that precise cycle-by-cycle synchronization can also be a feature of goal-directed odor-guided behavior. Mice selectively deploy this pattern of sniff-synchronized three-dimensional nose movement. For nose speed, yaw velocity, and Z-velocity, sniff synchrony is significantly reduced during the ITI when the mouse is returning from the reward port to initiate the next trial, even when the mouse is sniffing rapidly. Modulations in nose speed were slightly different than trial-shuffled data, showing that sniff-synchronized movement is not totally absent during the ITI, whereas modulations in yaw velocity and Z-velocity were indistinguishable from trial-shuffled data (*Figure 5—figure supplement 1*). This difference between within-trial and between-trial sniff synchrony was not contingent on the mouse's slower nose speed during the ITI (*Figure 5A*). Kinematic synchrony was the same when only periods of high-speed nose movement in the ITI are included in the analysis (*Figure 5—figure supplement 2*). This reduction of kinematic synchrony when the mouse is not performing the task suggests that sniff-synchronized movement is not an inevitable biomechanical accompaniment to fast sniffing, but rather reflects a strategic behavioral state. Further support for this idea comes from analyzing time intervals when the mouse attempts to initiate a trial before the end of the ITI.

After such premature attempts at trial initiation, the mice execute sniff-synchronized movement, despite the absence of the experimenter-applied odor stimulus (*Figure 5—figure supplement 2*). Lastly, sniff synchrony changes dramatically in the time interval between crossing the virtual decision line and entering the reward port, when odor is still present yet the animal has committed to a decision (*Figure 5—figure supplement 2*). Taken together, our observations indicate that sniff synchronous movement is a proactive, odor-seeking strategy rather than a reactive, odor-gated reflex.

## State space modeling finds recurring motifs that are sequenced diversely across mice

In our olfactory search paradigm, the overall rhythm of nose movement synchronizes with sniffing (*Figures 4* and *5*), and yet the mice move through a different trajectory on every trial (*Figure 3D*). Given this heterogeneity, it was not obvious to us how to best quantify common features of movement trajectories across trials and subjects. Rather than guess at suitable features, we used an unsupervised machine learning tool, modeling the movement data with an auto-regressive hidden Markov model (AR-HMM) (*Murphy, 2012*; *Poritz, 1982*). This model parses continuous sequential data into a discrete set of simpler movement motif sequences, similarly to 'Motion Sequencing' (MoSeq) (*Wiltschko et al., 2015*). We fit AR-HMMs to the allocentric three-point coordinate data (front of snout, back of head, and center of mass *Video 4*) pooled across a subset of mice and trial conditions (see Materials and methods and *Figure 2*; e.g., 80:20, 90:30, nostril stitch). Models were then tested for their ability to explain a separate set of held-out trials (see Materials and methods). These models defined discrete movement patterns, or 'motifs', that recur throughout our dataset (e.g., *Figure 6A*). We fit different AR-HMMs each constrained to find a particular number of motifs (between 6 and 100) and found that the cross-validated log-likelihood of these fits continued to rise up to 100 motifs (*Figure 6—figure supplement 1*). For visualization, we will focus on a model with 16 states, which we narrow to 11, by excluding rare motifs that take up <5% of the assigned video frames (*Figure 6B* and *Figure 6—figure supplement 1C, D*). Models with more or fewer states gave equivalent results (*Figure 6—figure supplements 2–4*).

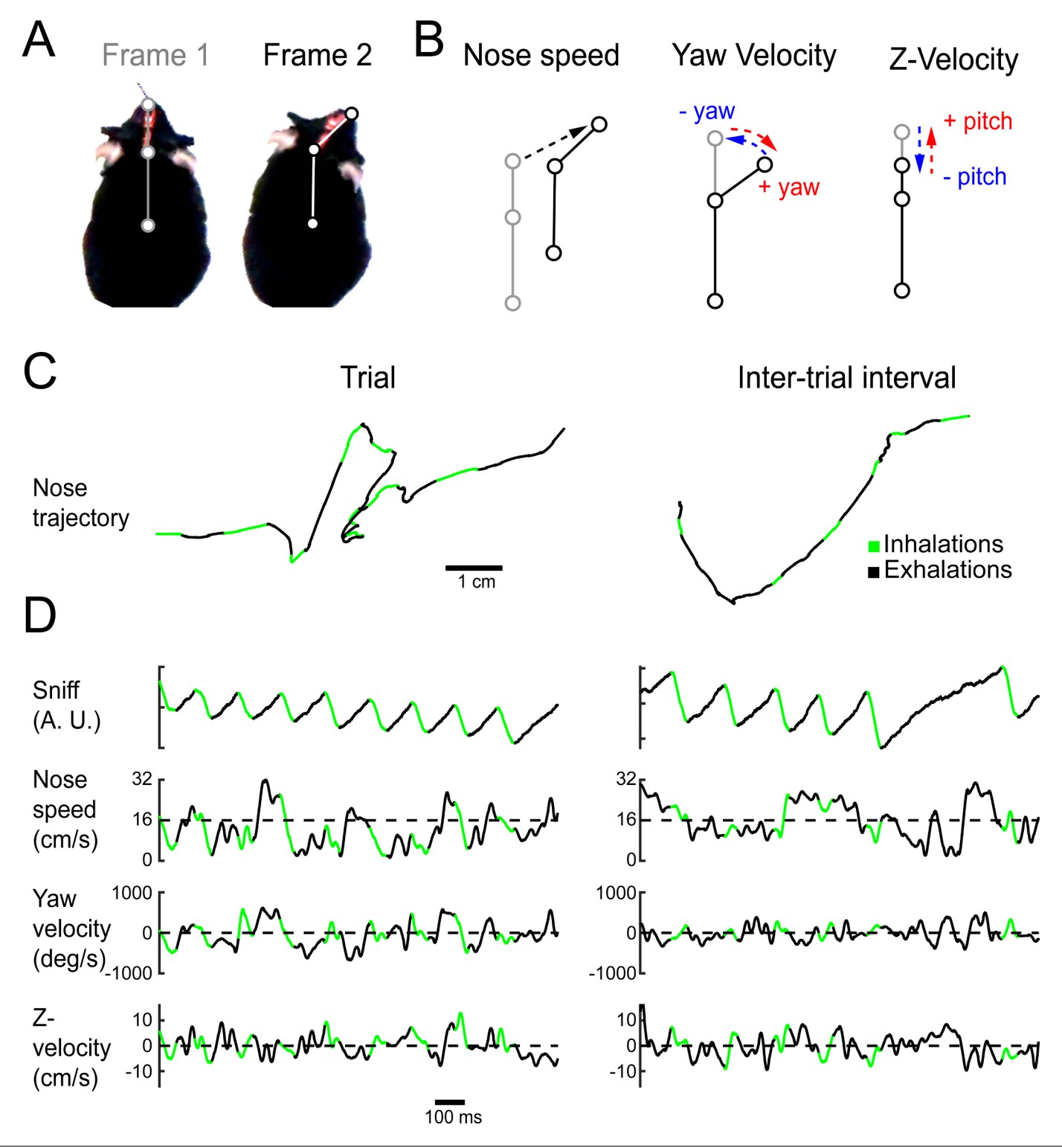

**Figure 4.** Quantifying kinematic parameters during olfactory search. (**A**) Schematic of kinematic parameters. Left: two example frames from one mouse, with the three tracked points marked: tip of snout, back of head, and center of mass. (**B**) Quantified kinematic parameters: 'nose speed': displacement of the tip of the snout per frame (12.5 ms inter-frame interval). 'Yaw velocity': change in angle between the line segment connecting snout and head and the line segment connecting head and center of mass. Centrifugal movement is positive, centripetal movement is negative. 'Z-velocity': change in distance between tip of snout and back of head. Note that this measure confounds pitch angle and Z-axis translational movements. (**C**) Segments of example trajectories. Left: the trajectory of the nose during 1 s of trial time. Green: path during inhalations. Black: path during the rest of the sniff.

*Figure 4 continued on next page*

*Figure 4 continued*

Right: same for an inter-trial interval trajectory. (D) Traces of sniff and kinematic parameters during the time windows shown in (C). Color scheme as in (C).

The motifs extracted by this model have interpretable spatiotemporal trajectories on average (*Figure 6B*, *Video 5*), although averaging masks considerable across-instance variability (*Video 6*). Across trials for a given mouse, motifs occurred in consistent but non-stereotyped sequences (*Figure 6C*, *Video 5*). Across mice, the model identified consistent behavioral features as motifs (*Figure 6—figure supplement 5*), but most mice were uniquely identifiable from how they sequenced motifs across trials. A classifier trained to decode mouse identity from the motif sequences on a trial-by-trial basis was able to perform above chance for eight out of nine mice (*Figure 6D*; p<0.01). Across the different concentration ratios (*Figure 2D*), movement sequences were not statistically distinguishable (*Figure 6E*). The only condition that gave distinguishable motif patterns were the odor omission trials (0:0), in which the mice made longer, more tortuous trajectories (*Figure 2C*). Thus, although this model is sensitive enough to decode mouse identity (*Figure 6D*), it does not detect stimulus-dependent modifications of sampling behavior, suggesting that the mice do not modify their sampling behavior in a gradient-dependent manner, at least in the movement parameters we measured. This lack of modification ran counter to our expectations because we reasoned that making the task harder would make the mice adjust their strategy to maintain high performance. We speculate that this absence of an adaptive strategy is due to impulsivity (*Miyazaki et al., 2012*; *Fonseca et al., 2015*).

## Movement motifs reveal two-state organization of olfactory search

Many behaviors have hierarchical structure that is organized at multiple temporal scales. Brief movements are grouped into progressively longer modules and are ultimately assembled into purposive behavioral programs (*Berman et al., 2016*; *Gallistel, 1982*; *Tolman, 1932*; *Weiss, 1968*).

Olfactory search programs in smaller organisms are often organized into two overarching states: move straight when concentration is increasing and reorient when concentration is decreasing (*Bargmann, 2006*; *Berg, 2000*; *Gomez-Marin et al., 2011*; *Kennedy and Marsh, 1974*; *Lockery, 2011*; *van Breugel and Dickinson, 2014*; *Vickers and Baker, 1994*). We hypothesized that olfactory search motifs in mice are organized similarly. To reveal higher-order structure in the temporal organization of these motifs, we applied a clustering algorithm that minimizes the Euclidean distance between rows of the Markov transition matrix (i.e., purely based on the conditional probabilities of motifs following them). This clustering separated motifs into two groups (*Figure 7A*), with several distinct properties. These properties were present in models with more or fewer states (*Figure 6—figure supplements 2–4*). Based on these differences (see below), we label these groups as putative 'investigation' and 'approach' states. First, investigation and approach motifs cluster their onset times in the trial, with investigation motifs tending to occur early in the trial, while approach motifs tend to begin later (*Figure 7B*). Grouping motifs into these higher-order states shows a consistent trial sequence, with trials beginning with investigation and ending with approach (*Figure 7C, D*). Importantly, entering the approach state is not a final, ballistic commitment to a given water port – switches from approach back to investigation were common (*Figure 7C, D*, *Video 7*). This pattern suggests that the mice are continuously integrating evidence about the odor gradient throughout their trajectory to the target. Second, these states correlated with distinct sniff rates and movement speeds. During investigation motifs, the mice moved more slowly and sniffed more rapidly, whereas the approach states were associated with faster movement and slower sniffing (*Figure 7E*) (*Video 8*). Third, the sniff-synchronized kinematic rhythms (*Figures 4* and *5*) were distinct in the two states (*Figure 7F*; Kolmogorov–Smirnov test, p<0.01). Specifically, nose speed and yaw velocity are more synchronized with sniffing during the investigation state (*Figure 7F*). Given the consistent sequence from investigation to approach and given that mice sniff faster during the early part of trial, these differences in kinematic parameters could reflect across-trial tendencies instead of within-trial synchrony. To test this possibility, we calculated the Kolmogorov–Smirnov statistic, which quantifies the difference between two cumulative distributions, for real and trial-shuffled data (*Figure 8—figure supplement 1*). This analysis showed that nose speed and yaw velocity modulation exceeded what

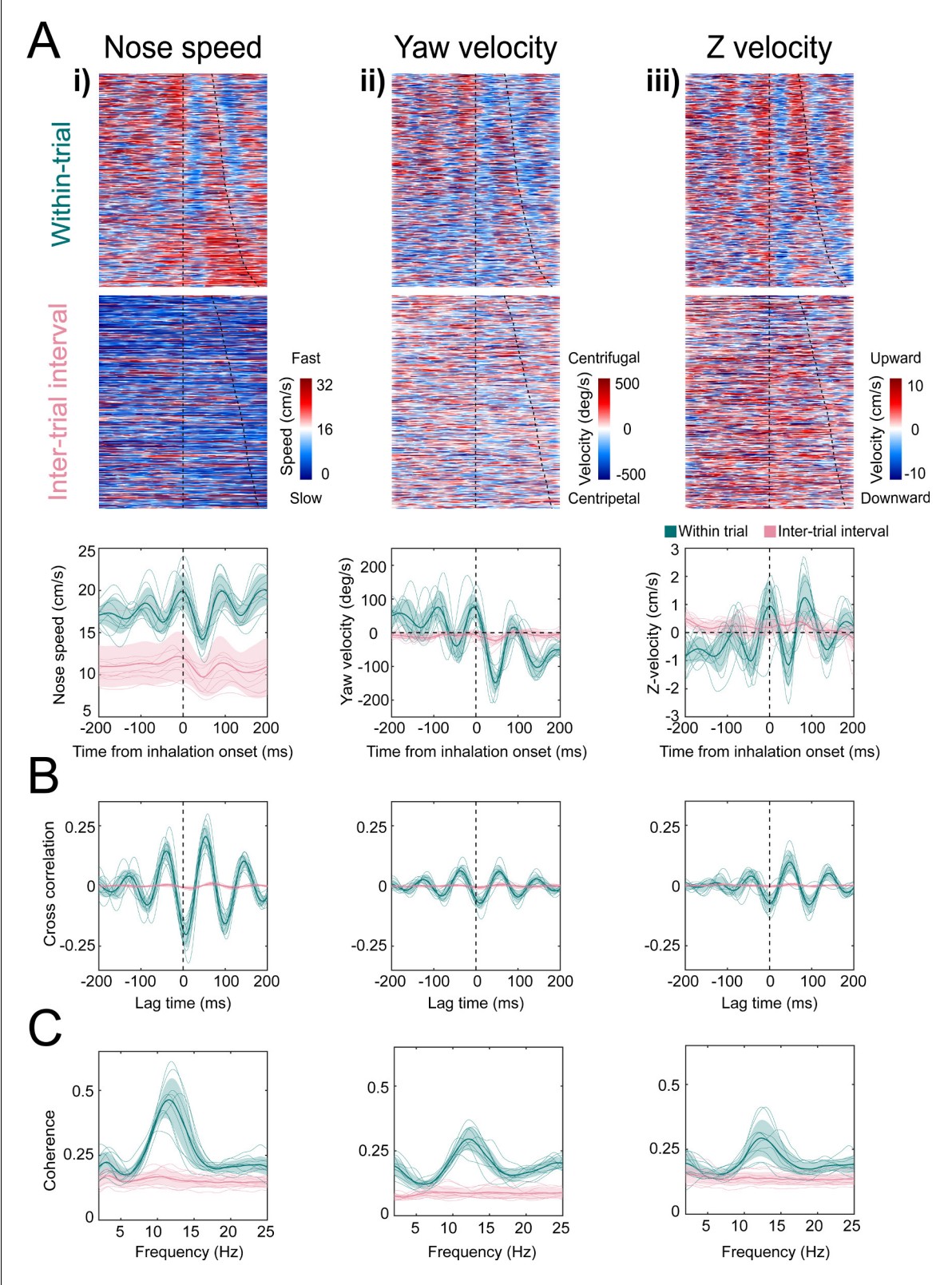

**Figure 5.** Kinematic rhythms synchronize with the sniff cycle selectively during olfactory search. (i–iii) Nose speed, yaw velocity, and Z-velocity, respectively (see *Figure 4* for definitions). (**A**) Top: color plot showing movement parameter aligned to inhalation onset for within-trial sniffs taken before crossing the decision line. Taken from one mouse, one behavioral session. Dotted line at time 0 shows inhalation onset, the second line demarcates the end of the sniff cycle, sorted by duration. Data are taken from one behavioral session. Middle: color plot showing each movement

*Figure 5 continued on next page*

*Figure 5 continued*

parameter aligned to inhalation onset for inter-trial interval sniffs taken before the first attempt at premature trial initiation. Bottom: sniff-aligned average of each movement parameter. Thin lines represent individual mice (*n* = 11), bolded lines and shaded regions represent the grand mean ± standard deviation. Green: within-trial sniffs; pink: inter-trial interval sniffs. (B) Normalized cross-correlation between movement parameter and sniff signal for the same sniffs as above. (C) Spectral coherence of movement parameter and sniff signal for the same sniffs as above.

The online version of this article includes the following figure supplement(s) for figure 5:

**Figure supplement 1.** Sniff synchronization shuffle test.

**Figure supplement 2.** Kinematic rhythms for premature initiations during the inter-trial interval and between decision line and reward port during trials.

would be expected from across-trial tendencies (1000 shuffles, p<0.001), while the Z-velocity modulation did not (p=0.31). Switches between the investigation and approach state mark behavioral inflection points that can be identified from trial to trial. We reason that these behavioral inflection points are a signature of key moments in the mouse's evolving decision process. Thus, our analysis can provide a framework for temporal alignment of diverse movement trajectories with simultaneously recorded physiological data (*Markowitz et al., 2018*).

## Investigation motif onsets are precisely locked to sniffing

If motif transitions correspond to relevant behavioral events, their temporal structure should correlate with the temporal structure of neural activity (*Markowitz et al., 2018*). During fast sniffing, respiration matches with the rhythms of head movement (*Figures 4* and *5*), whisking, and nose twitches (*Kurnikova et al., 2017*; *Moore et al., 2013*; *Ranade et al., 2013*). These motor rhythms correlate with activity in numerous brain regions, including brainstem, olfactory structures, hippocampus, amygdala, and numerous neocortical regions (*Karalis and Sirota, 2018*; *Kay, 2005*; *Macrides et al., 1982*; *Vanderwolf, 1992*; *Yanovsky et al., 2014*; *Zelano et al., 2016*). We hypothesized that movement motifs would lock with these behavioral and neural rhythms, so we aligned sniff signals with motif onset times. Importantly, the breath signal was not input to the model.

This alignment revealed a striking organization of motif sequences relative to the sniff rhythm. For example, the onset times of motif 6 (dark blue) occurred in a precise timing relationship with sniffing (*Figure 8A*). To visualize the timing relationship between onsets of all motifs and sniffing, we calculated the equivalent of a peristimulus time histogram for inhalation times relative to the onset time of each motif and took the grand mean across all mice (*Figure 8B*; *n* = 4). Further, to determine how motif onset times are organized relative to the sniff cycle, for each motif we calculated a histogram of motif onset in sniff phase coordinates (*Figure 8C*; relative position in the sniff cycle). Sharp peaks are apparent in both histograms for investigation motifs, and less so for approach motifs (quantified below; *Figure 8B, C*). Importantly, these timing relationships are consistent across mice, with some motifs tending to occur early in the sniff cycle during inhalation and others occurring later in the sniff cycle (*Figure 8D*). Thus, parsing diverse movement trajectories into sequences of recurring movement motifs reveals additional sniff-synchronized kinematic structure in a consistent manner across mice.

Are motif onsets timed with respect to inhalation times, or do they coordinate with the entire sniff cycle? In other words, is motif onset probability more modulated in time or phase? To quantify the sniff synchronization of motif onset times, we calculated a modulation index $(MI = (max - min)/(max + min))$ for each motif's across-mouse mean histogram (*n* = 4). To test whether these trial-by-trial modulation indices exceeded what would be expected from across-trial tendencies, we compared real and trial-shuffled data (*Figure 8—figure supplement 2*). All investigation motifs were significantly modulated for both time and phase coordinates (*Figure 8E*; filled symbols, permutation test, p<0.001), with some having higher *MI* in time, and others in phase. One approach motif was significantly modulated in time coordinates (*Figure 8E*; right-half filled symbol, p=0.003), while two approach motifs were significantly modulated in phase coordinates (*Figure 8E*; left-half filled symbols, p=0.015 and p<0.001). Comparing the modulation indices between time and phase coordinates does not reveal a consistent pattern of modulation in time vs. phase – some motifs had higher *MI* in phase, others in time. Thus, our data are inconclusive as to how motif onsets organize relative to the sniff cycle. Nevertheless, these analyses demonstrate that kinematic inflection points synchronize with breathing during olfactory search. Given that breathing synchronizes to

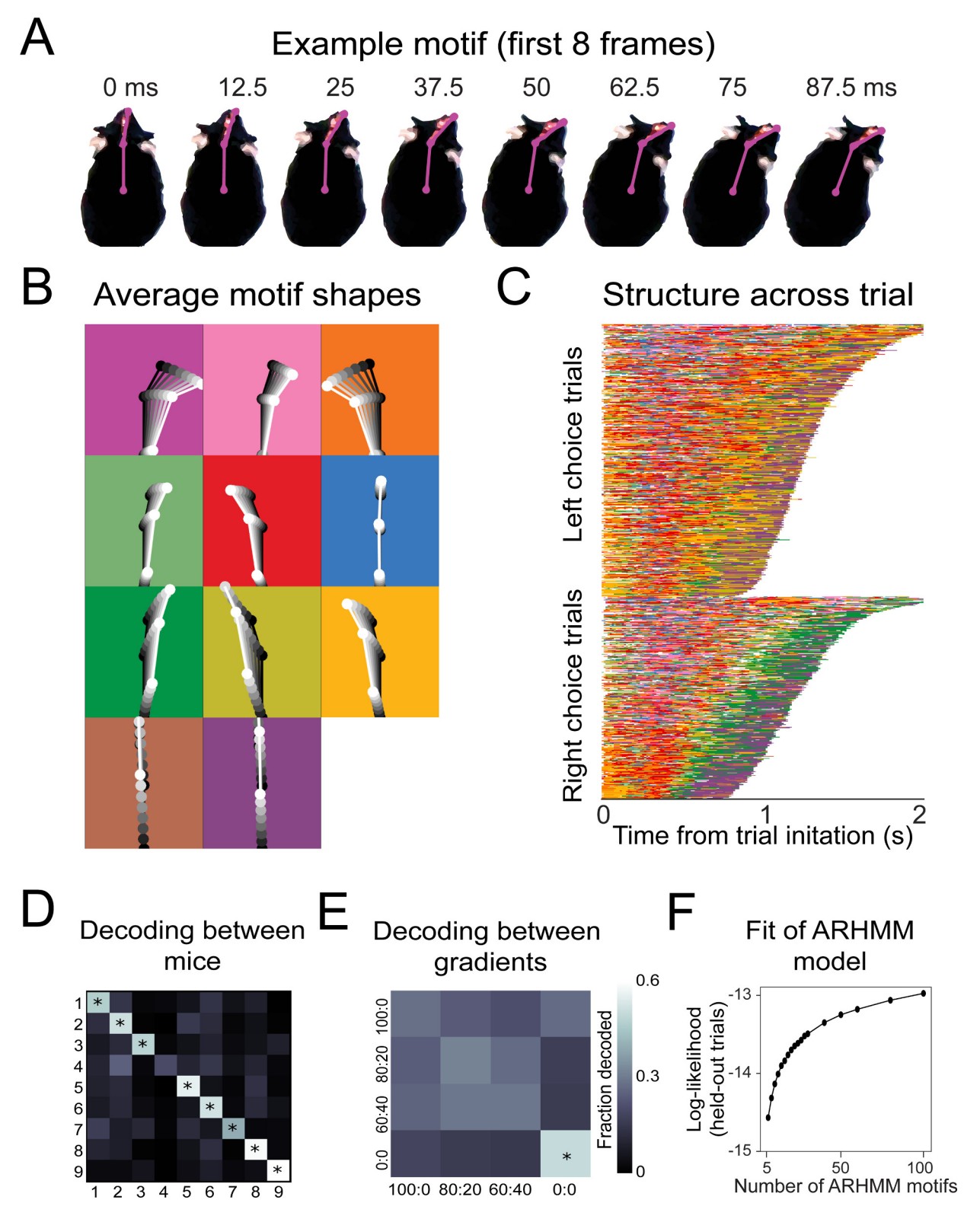

**Figure 6.** Recurring movement motifs are sequenced diversely across mice and consistently across stimuli. (**A**) Eight example frames from one instance of a behavioral motif with tracking overlaid. (**B**) Average motif shapes. Dots and lines show the average time course of posture for eight frames of each of the 11 motifs ($n$ = 9 mice). All instances of each motif are translated and rotated so that the head is centered and the head-body axis is oriented upward in the first frame. Subsequent frames of each instance are translated and rotated the same as the first frame. Time is indicated by color (dark to

*Figure 6 continued on next page*

*Figure 6 continued*

light). Background color in each panel shows the color assigned to each motif. (C) Across-trial motif sequences for two behavioral sessions for one mouse. Trials are separated into trials where the mouse chose left and those in which the mouse chose right. Trials are sorted by duration. Both correct and incorrect trials are included. Color scheme as in (B). (D) Linear classifier analysis shows that mice can be identified from motif sequences on a trial-by-trial basis. Grayscale represents the fraction of trials from a given mouse (rows) that are decoded as belonging the data of a given mouse (columns). The diagonal cells represent the accuracy with which the decoded label matched the true label, while off-diagonal cells represent trials that were mislabeled by the classifier. Probabilities along rows sum to 1. Cells marked with asterisks indicate above chance performance (label permutation test, p<0.01). (E) Linear classifier analysis identifies odor omission trials above chance, but does not discriminate across odor concentration ratios (*n* = 9 mice). (F) Cross-validated log-likelihood (evaluated on trials not used for model fitting) for fit auto-regressive hidden Markov model (AR-HMM) models with different numbers of motifs, *S*, shows that model log-likelihood does not peak or plateau up to *S* = 100.

The online version of this article includes the following figure supplement(s) for figure 6:

**Figure supplement 1.** Motif statistics and examples and linear decoder results for 80:20 experiments.

**Figure supplement 2.** Motif shapes, sequences, transition matrices, and sniff synchronization for an auto-regressive hidden Markov model capped at a maximum of six states.

**Figure supplement 3.** Motif shapes, sequences, transition matrices, and sniff synchronization for an auto-regressive hidden Markov model capped at a maximum of 10 states.

**Figure supplement 4.** Motif shapes, sequences, transition matrices, and sniff synchronization for an auto-regressive hidden Markov model capped at a maximum of 20 states.

**Figure supplement 5.** Motif shapes across individuals.

other motor and brain rhythms, these motifs likewise correlate to the structure of activity of many neurons. Thus, our analysis will be a useful tool to pinpoint behaviorally relevant activity in widespread brain regions.

## Investigation and approach occupancy maps suggest a serial-sniff comparison strategy

We propose that motif transition times indicate 'decision points' at which the animal chooses its next move (*Markowitz et al., 2018*). The transitions between investigation and approach motifs are particularly relevant since investigation motifs may correspond to an evidence-gathering state, while approach motifs may correspond to a reward-gathering state. What kind of sensory evidence guides transitions between investigation and approach? Although we cannot determine the precise odor inputs the mice acquire on a sniff-by-sniff basis, we reasoned that we could elucidate the search strategy by examining aggregate across-trial patterns in allocentric maps of investigation and approach occupancy.

As expected from the temporal structure of investigation and approach (*Figure 7C, D*), the mice primarily investigate near the initiation port (*Figure 9A*; *n* = 9 mice) and primarily approach close to the decision line (*Figure 9A*, orange). Along the longitudinal axis of the arena, the two occupancy maps overlap in a region between initiation port and decision line (*Figure 9A*, black) where overall occupancy also peaks (*Figure 3F*). Along the lateral axis of the arena, on average the overlap region is roughly centered on the lateral midline between left and right sides (*Figure 9A*). However, this centered position is not representative of the individual mice, which have their overlap region in different positions relative to the lateral midline, with some on the left, and others on the right (*Figure 9—figure*

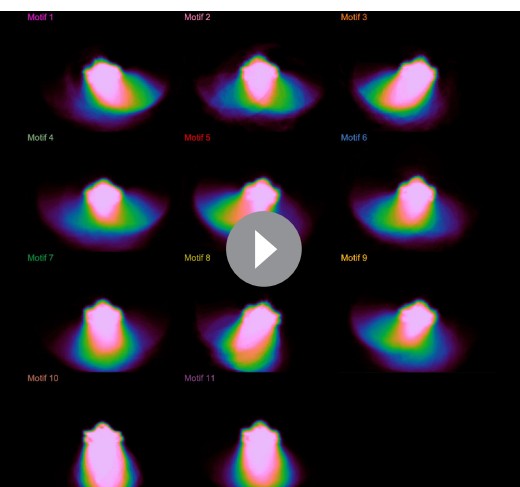

**Video 5.** Moving occupancy histograms for motifs show their average movement dynamics. We aligned every instance of a given motif such that that instance's frames were translated to position the center of mass in frame 1 at consistent location in the image and rotated so that the body axis points upward in frame 1. Colormap represents regions of high occupancy with brighter, warmer colors, and lower occupancies with darker, colder colors. Video frame rate is slowed by 8×.
https://elifesciences.org/articles/58523#video5

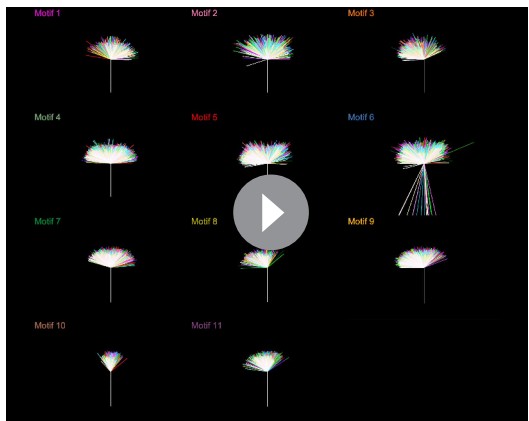

**Video 6.** Moving wireframes for motifs show the variability of movement dynamics for a given motif. Wireframes consist of two lines connecting coordinates of front of snout, back of head, and center of mass extracted using Deeplabcut. Each wireframe represents a single instance of every motif. We aligned frames as in *Video 5*. Lines are colored arbitrarily to facilitate visualization of individual wireframes. Video frame rate is slowed by 8×.
https://elifesciences.org/articles/58523#video6

*supplement 1A*). However, if trials are oriented such that the chosen side is always up in the occupancy maps, the overlap region is displaced toward the chosen side of the arena in all individual mice (*Figure 9—figure supplement 1B*). Thus, the mice primarily switched states while located on the side they would ultimately choose. To quantify the overlap between states, we calculated a relative occupancy index, defined as the difference in investigation and approach occupancy divided by their sum (*Figure 9C*, I.A.I.). For this index, a bin where the mice primarily investigated has a positive value, while a bin primarily occupied during the approach state has a negative value. Along the longitudinal axis, most of the change in this index occurred between inflection points at 5 and 10 cm, which we define as a 'transition zone' for the analyses below (*Figure 9D*).

Along the lateral axis of the arena, I.A.I. was quite variably distributed across mice, both for the entire occupancy map and within the transition zone (*Figure 9E*), consistent with the individual mouse occupancy maps (*Figure 9—figure supplement 1A*). Orienting trials with respect to the chosen side demonstrates a clearer pattern, with primarily investigation on the unchosen side and primarily approach on the chosen side (*Figure 9F* and *Figure 9—figure supplement 1B*).

Occupancy maps allowed us to further evaluate hypotheses about the search strategy mice use in these conditions. One hypothetical strategy is that the mice memorize absolute concentrations across trials and compare each individual sniff to an internal threshold learned over previous trials (single-sniff hypothesis). Another possible strategy would be serial-sniff comparison, where the mouse senses changes between sequential samples within individual trials (serial-sniff hypothesis).

These hypotheses make distinct predictions about where the mouse should sample. For the single-sniff hypothesis, the most informative location to sample is directly downwind of the odor ports, where concentration differences between left and right trials are maximal (*Figure 1—figure supplement 2*). For gradient sensing, the optimal location is instead across the lateral midline, where the gradients are sharpest (*Figure 1—figure supplement 2*; *Yovel et al., 2010*). We tested these predictions by comparing occupancy maps between correct and incorrect trials. For the single-sniff strategy, the mouse should get it correct more often when it investigates downwind of the odor ports, while a serial sniff hypothesis predicts that correct trials should show increased investigation at the midline.

Correct and incorrect trials yielded qualitatively similar occupancy maps (*Figure 10A*). To quantify their differences, we first compared their occupancy indices along the longitudinal axis of the arena (*Figure 10B, C*). Correct trials featured significantly higher I.A.I. (greater investigation) in

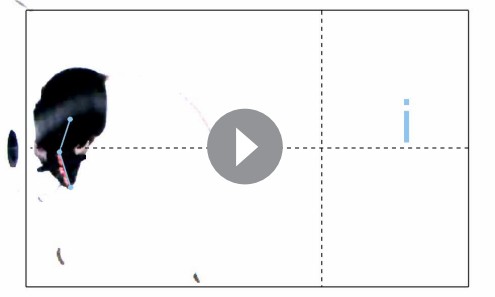

**Video 7.** Example trials with investigation/approach overlaid. Three dots on the mouse represent the coordinates of front of snout, back of head, and center of mass extracted using Deeplabcut. Dots and lines are colored according to whether that frame was assigned by the auto-regressive hidden Markov model to an investigation motif or an approach motif. Sniffing is indicated by sound (higher tone = inhalation, lower tone = exhalation). Video frame rate is slowed by 8×.
https://elifesciences.org/articles/58523#video7

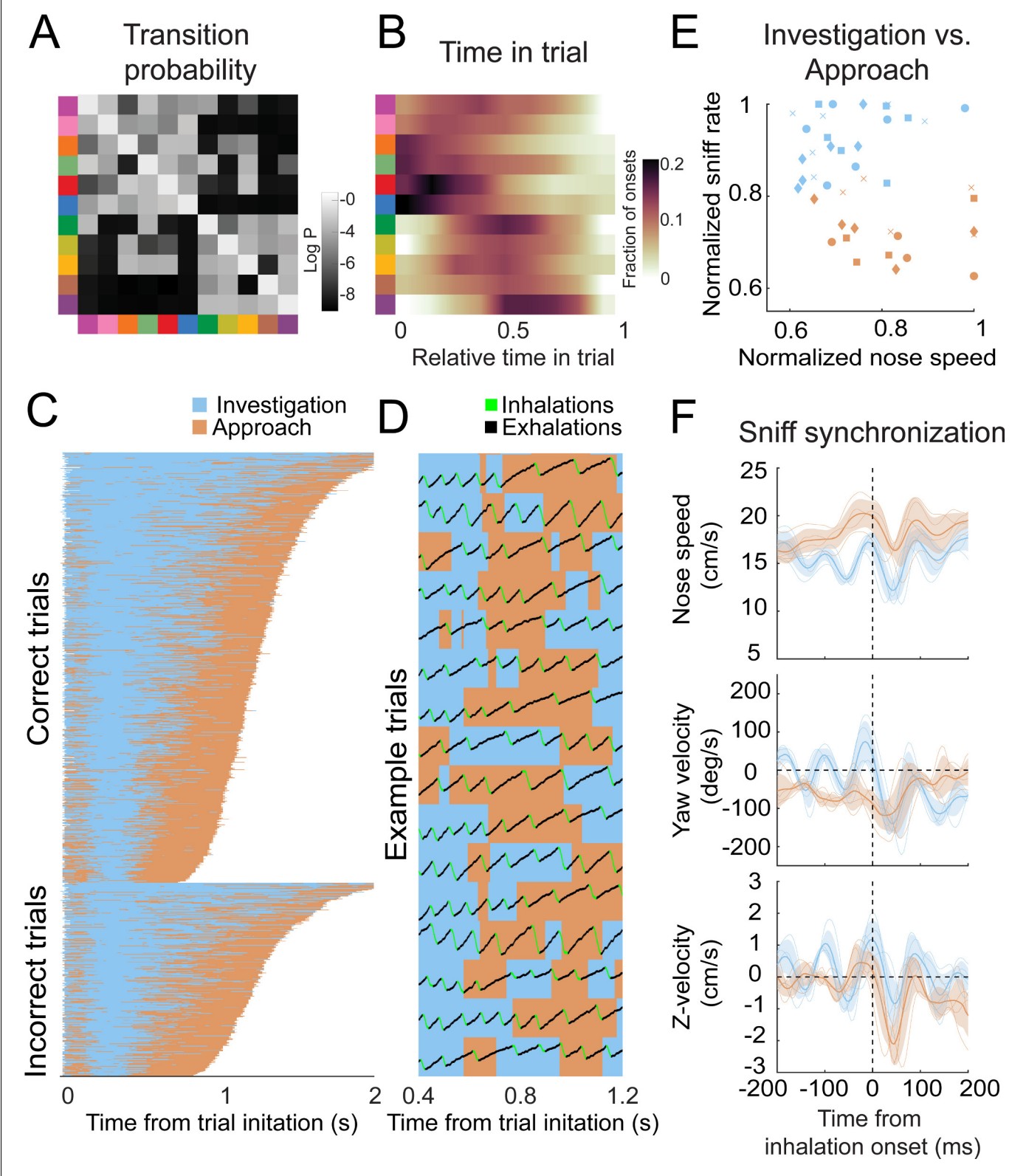

**Figure 7.** Behavioral motifs can be categorized into two distinct groups, which we putatively label as investigation (blue) and approach motifs (orange). Colors in panel A & B refer to motifs specified in *Figure 6*. (**A**) Transition probability matrix. Grayscale represents the log probability with which a given motif (rows) will be followed by another (columns). Clustering by minimizing Euclidean distance between rows reveals two distinct blocks of motifs. We label the top-left block as 'investigation' and the bottom-right block as 'approach'. (**B**) Distribution of onset times for each motif, normalized by trial

*Figure 7 continued on next page*

*Figure 7 continued*

duration. Investigation motifs tend to occur early in trials, while approach motifs tend to occur later (*n* = 9 mice). (**C**) Across-trial motif sequences for two behavioral sessions for one mouse, with motifs classified into investigation and approach. Trials are separated into correct trials (above) and incorrect trials (below). Motif sequences are sourced from the same data as *Figure 6C*. (**D**) Temporal details of investigation-approach transitions with overlaid sniff signal. Data come from a subset of trials shown in (**C**). In the sniff signal, green represents inhalations, black represents the rest of the sniff. (**E**) Investigation and approach motifs differ in nose speed and sniff rate. Individual markers represent one motif from one mouse. Marker shapes correspond to the individual mice (*n* = 4). Sniff rate and nose speed are normalized within mice. (**F**) Investigation and approach motifs differ in the kinematic rhythms (same parameters as in *Figures 4* and *5*). Thin lines represent individual mice (*n* = 4), thick lines and shaded regions represent the grand mean ± standard deviation. Blue: within-trial sniffs; orange: inter-trial interval sniffs. Top: nose speed modulation, defined by a modulation index ($maxspeed - minspeed$)/($max + min$) calculated from the grand mean, is significantly greater for investigation motifs than approach motifs (*Figure 8—figure supplement 1*; p<0.001, permutation test). Middle: yaw velocity modulation is significantly greater for investigation motifs than approach motifs (*Figure 8—figure supplement 1*; p<0.001, permutation test). Bottom: Z-velocity modulation does not significantly differ between approach motifs and investigation motifs (p=0.31, permutation test).

the latter part of the transition zone, while past the decision zone the I.A.I. was higher for incorrect trials (*Figure 10C*; permutation test, p<0.001, *n* = 9 mice). Thus, increased investigation in the transition zone was associated with correct trials, while increased investigation near the decision line was associated with incorrect trials. This pattern suggests that investigation is not inherently advantageous to olfactory search irrespective of location. Instead, it matters where the mouse investigates, and some locations are less advantageous. Notably, absolute concentrations are most discriminable near the decision line (*Figure 1—figure supplement 2C*), suggesting that mice may not be able to capitalize on this cue under these conditions.

We next quantified state occupancies along the lateral axis within the transition zone (*Figure 10D–F*). Correct trials featured significantly increased investigation at and on the unchosen side of the midline relative to incorrect trials (*Figure 10G*; permutation test, p<0.001, *n* = 9 mice). By definition, occupancy of the unchosen side precedes a crossing of the midline to get to the chosen side. This suggests an advantage to sampling both sides of the midline, consistent with a serial-sniff gradient sensing strategy. Further, investigation more laterally, downwind of the odor port, was increased on incorrect trials, suggesting that sampling this location was not advantageous for task performance, contrary to the single-sniff absolute concentration hypothesis. Approach occupancy showed a different pattern, with significantly higher approach at and around the midline on incorrect trials, and a significant increase in approach occupancy closer to the chosen water port (*Figure 10H*; permutation test, p<0.001). Consistent with these observations, on correct trials I.A.I. showed significant elevation at the midline and into the unchosen side of the arena, while increased I.A.I. of the chosen side was associated with incorrect trials (*Figure 10I*; permutation test, p<0.001). Altogether, these results suggest that it is advantageous to sample both sides of the midline in this task, consistent with the serial-sniff hypothesis.

An important consideration in interpreting these results pertains to the construction of our task. Before choosing a side, the mice have to turn out of the initiation port in one direction or the other on every trial. On some trials they stay and choose the side of the first turn, while on other trials they switch and choose the other side. A single-sniff hypothesis predicts that if the mouse happens to turn first toward the correct side, it will tend to encounter above threshold concentrations during the turn and should therefore tend to transition to approach without crossing the midline. However, if the mouse turns first to the incorrect side, threshold crossings will tend not to occur and the mouse can initiate approach before crossing the midline.

Thus, this hypothesis predicts that correct vs. incorrect occupancy differences should occur at

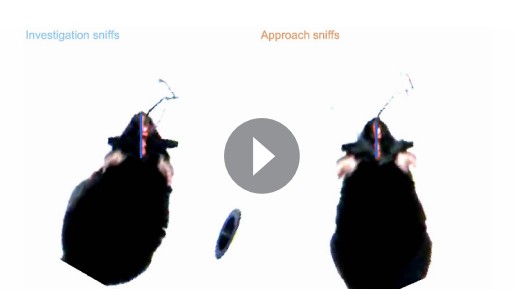

**Video 8.** Movement trajectories for individual sniffs separated into investigation and approach. Each video snippet corresponds to one sniff, where the frames are translated so that the back of the head is centered and rotated so that the head angle is vertical, in the first frame of each sniff. Blue = inhalation, pink = exhalation. Video frame rate is slowed by 10×.
https://elifesciences.org/articles/58523#video8

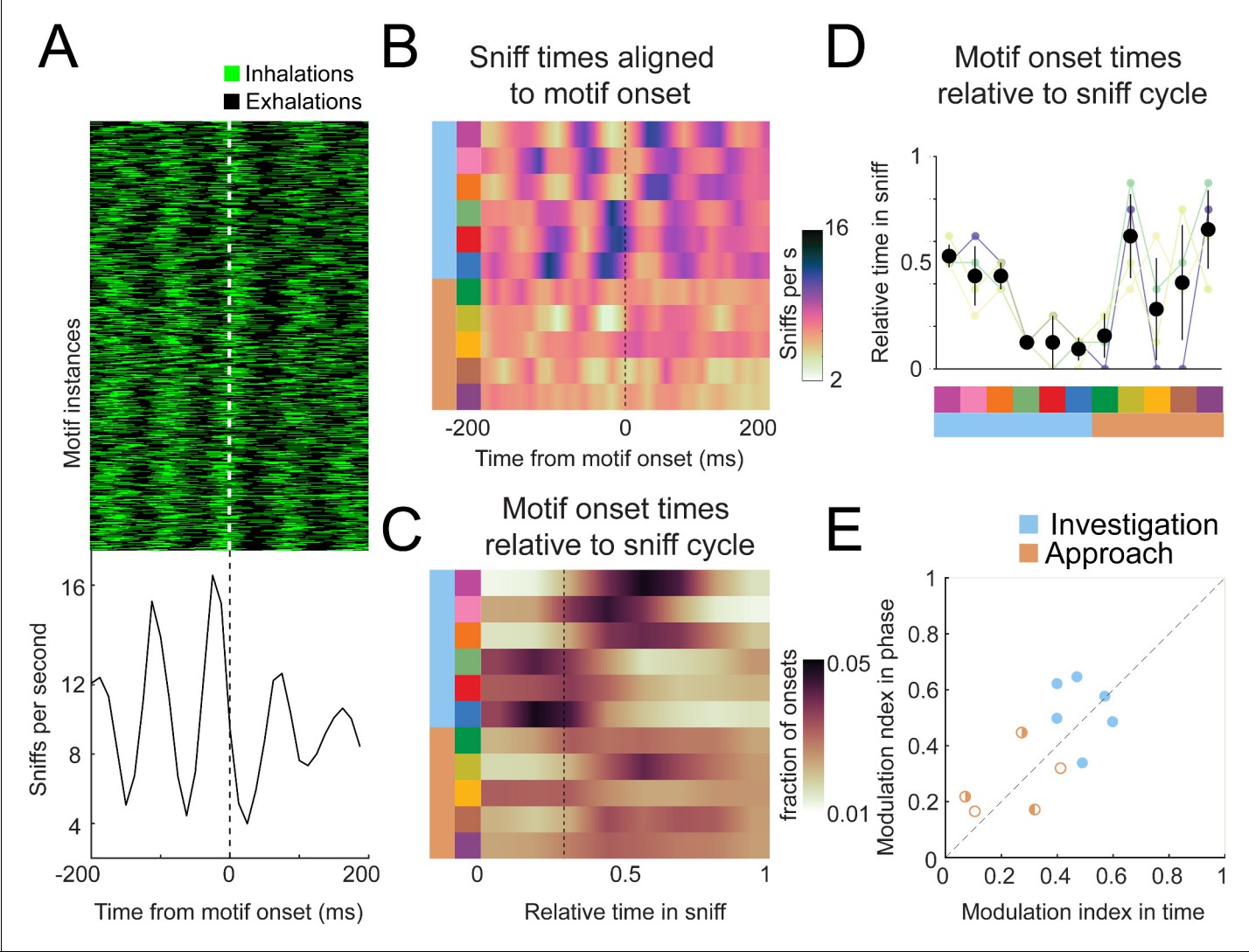

**Figure 8.** Motif onsets synchronize to the sniff cycle. (A) Alignment of the sniff signal to an example motif. Top: color scheme shows sniff cycles aligned to the onsets of motif 6 (blue). Motif instances are in chronological order. Green: inhalation; black: rest of sniff. Bottom: peristimulus time histogram of inhalation times aligned to the onset of motif 6. (B) Alignment of sniff signal to onset times of all motifs across mice (n = 4). Motifs categorized into two types we call investigation (light blue) and approach (orange). Colormap represents the grand means for peristimulus time histograms of inhalation times aligned to the onset of motifs. (C) Alignment of motif onset times in sniff phase. Colormap represents peristimulus time histograms of motif onsets (bin width = 12.5 ms) times aligned to inhalation onset, with all sniff durations normalized to 1. Dotted line shows the mean phase of the end of inhalation. (D) Motif alignment to sniff phase is consistent across mice. Thin lines represent individual mice, black points are means, and whiskers are ±1 standard deviation (n = 4 mice). (E) Investigation motifs are more synchronized to the sniff cycle than approach motifs. Dots represent the modulation index in time on the x-coordinates and in phase on the y-coordinates. Filled dots represent motifs that are significantly modulated in both time and phase (p<0.01, permutation test). Half-filled dots represent motifs that are significantly modulated in time (left half filled) or phase (right half filled). The online version of this article includes the following figure supplement(s) for figure 8:

**Figure supplement 1.** Shuffle test for the difference in sniff synchronization between investigation and approach motifs for movement parameters.
**Figure supplement 2.** Shuffle test for sniff synchronization of motif onset for investigation and approach motifs.

different positions along the lateral axis for stay and switch trials. To test this prediction, we performed the same analyses separately for stay and switch trials. Although not identical, both stay and switch trials showed significantly increased investigation at and on the unchosen side of the midline for correct trials (*Figure 10—figure supplement 1*). This analysis demonstrates that the apparent advantage of sampling across the midline is not an artifact of the asymmetry between switch and stay trials. Taken together, investigation and approach occupancy mapping provides further

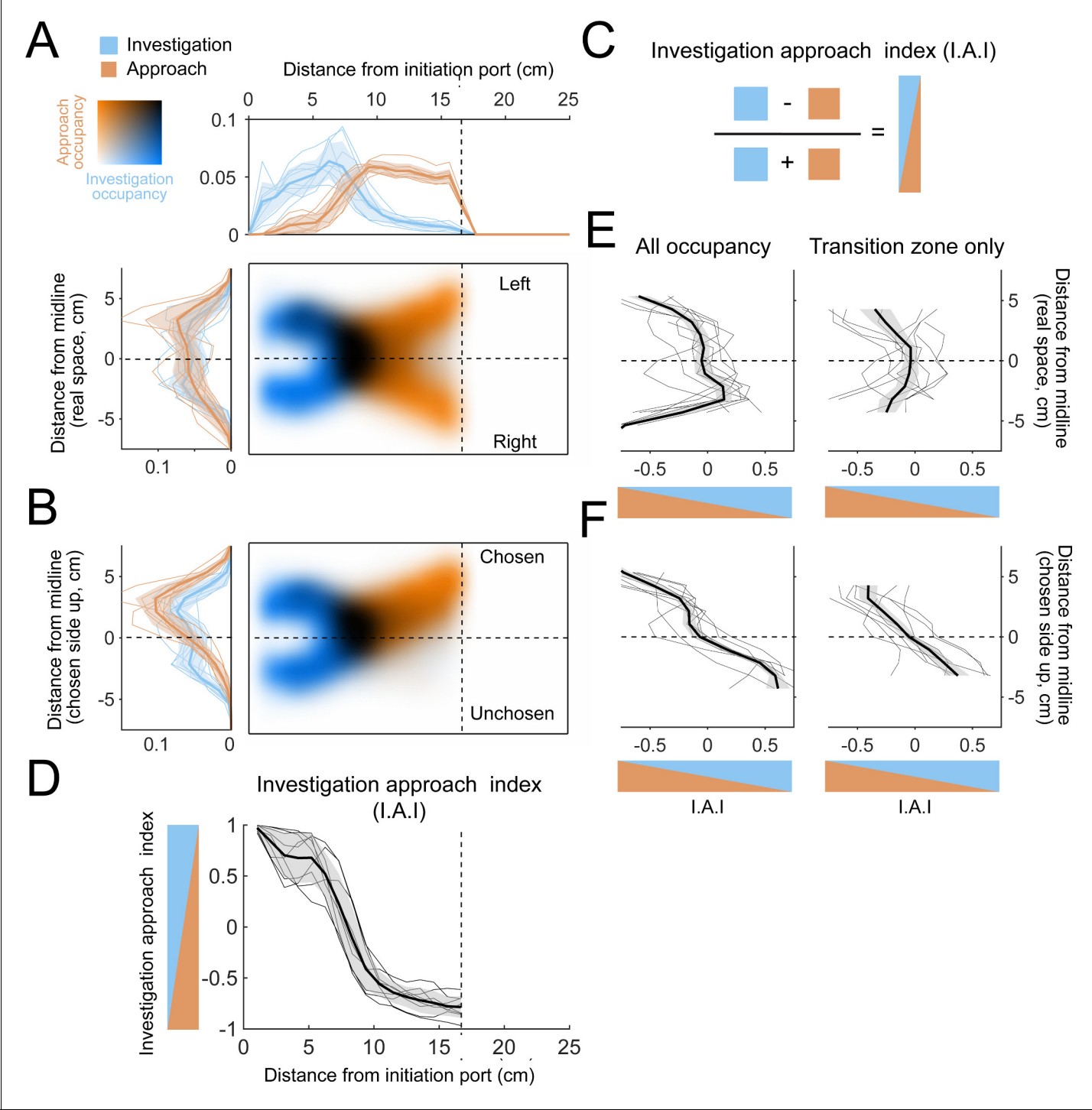

**Figure 9.** The allocentric spatial distribution of investigation and approach occupancy. (**A**) Colormaps show two-dimensional histograms of the occupancy density (1 cm$^2$ bins, $n$ = 9 mice) with investigation density in blue, approach density in orange, and overlap shown by darker coloring (key in top-left corner). Histograms around the colormaps show the state occupancy projected onto the longitudinal (top) and lateral (left) axes of the arena. (**B**) Occupancy distributions after the right-choice trials are flipped upward so that the chosen side is always facing up in the diagram. (**C**) Relative usage is quantified with an investigation approach index (I.A.I.), defined as the difference between investigation and approach occupancy divided by their sum. Blue and orange triangles are visual aids that represent the I.A.I. (**D**) Relative occupancy density of investigation and approach (I.A.I.), plotted along the longitudinal axis of the arena from the initiation port to the decision line. We define the region between 5 cm and 10 cm as a 'transition zone', in which most transitions between investigation and approach take place. Thin lines are individual mice ($n$ = 9), thick line and shaded region are mean ± s.e.m. (**E**) I.A.I. plotted along the lateral axis in real space (i.e., left-right orientation) for all occupancy throughout the arena (left) and for the

*Figure 9 continued on next page*

*Figure 9 continued*

transition zone only (right). Thin lines are individual mice (*n* = 9), thick line and shaded region are mean ± s.e.m. (**F**) Same as (**E**), but after the lateral axis has been reoriented so that the chosen side is always up.

The online version of this article includes the following figure supplement(s) for figure 9:

**Figure supplement 1.** The allocentric spatial distribution of investigation and approach occupancy for individual mice.

evidence, suggesting that mice use a serial-sniff strategy to sense gradient cues in this task (*Catania, 2013*).

## Discussion

This study elucidates sensory computations and movement strategies for olfactory search by freely moving mice. Mice learn our behavioral task in days, after which they perform approximately 150 trials daily, sometimes for months. Task performance worsens for shallower odor gradients at a fixed absolute concentration, but is unaffected by varying absolute concentrations at a fixed concentration gradient. Taken together, these results show that mice can navigate noisy gradients formed by turbulent odor plumes. This gradient-guided search is robust to perturbations including novel odorant introduction and naris occlusion. These results give insight into sensory computations for olfactory search and constrain the possible underlying neural mechanisms.

Mice perform this task with a strategic behavioral program. During search, mice synchronize rapid three-dimensional head movements with fast sniffing. This synchrony is not a default accompaniment of fast sniffing – synchrony is absent when the mice are not searching. Movement trajectories are not stereotyped, but vary considerably across trials. To manage this complexity, we took an unsupervised computational approach to parse heterogeneous trajectories into a small number of movement motifs that recur across trials and subjects. This analysis captures common movement features across mice, but individual mice can be identified by how they sequence these motifs. Our model was not constrained to find structure at a specific timescale, and consequently identified very brief, simple motifs. To find higher-order temporal structure in the data, we clustered motifs by their transition probabilities, which revealed two clear categories, putatively corresponding to investigation and approach. Investigation motifs tend to be executed early in the trial, and entail slower movement, faster sniffing, and more sniff synchrony than approach motifs. Even so, approach motifs are not ballistic commitments to an answer – switches from approach to investigation occurred on many trials. Lastly, the onset times of motifs were precisely locked to sniffing, with investigation motifs starting at characteristic phases of the sniff cycle.

The allocentric structure of investigation and approach suggests that the investigation state is not inherently advantageous. Rather, where the mouse investigates matters for performance. This dependence of performance on location indicates the spatial distribution of informative features in this olfactory scene. Notably, incorrect trials feature more investigation directly downwind of the odor source, along the axis of maximal odor concentration, which would be optimal if the mouse were using a single-sniff, absolute concentration strategy (*Figure 1—figure supplement 2C*). Thus, these analyses provide further evidence that the mice do not capitalize on absolute concentration information to guide performance in this task. Instead, correct trials feature more investigation at and across the axis of maximal odor gradient (*Figure 1—figure supplement 2D*), reminiscent of an object localization strategy observed in Egyptian fruit bats. When approaching an object, these bats do not center their sonar beams directly at the object, but rather point them off axis, so that the maximum slope of the acoustic profile intersects the object (*Yovel et al., 2010*). Likewise, in this task mice do not gain an advantage by centering their sniffing directly downwind of the odor sources, but rather perform best when they investigate the location of the steepest slope of the odor gradient, consistent with a serial-sniff gradient sensing strategy. Thus, our unsupervised computational analysis of airborne odor tracking supports the idea that sampling off axis can be an optimal strategy for localization across diverse sensory systems and species (*Yovel et al., 2010*).

Olfactory navigation can be either guided or gated by odor (*Baker et al., 2018*). Some organisms operate in a regime where diffusion forms smooth chemical gradients, in which classical chemotaxis strategies can be effective (*Bargmann, 2006*; *Berg, 2000*; *Gomez-Marin and Louis, 2012*;

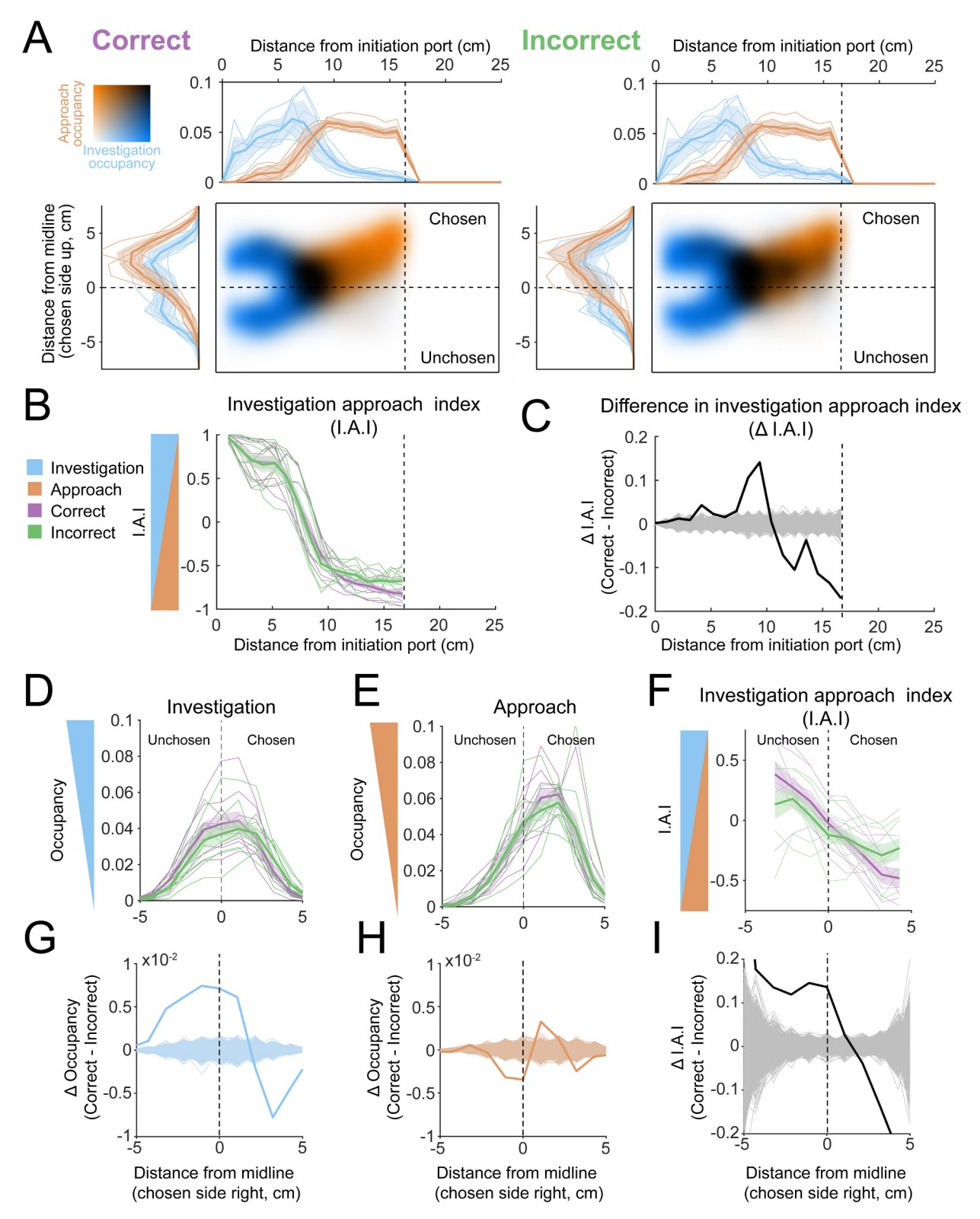

**Figure 10.** Occupancy maps indicate an advantage for investigation of both sides. (**A**) Colormaps show two-dimensional histograms of the occupancy density (1 cm² bins, *n* = 9 mice) with investigation density in blue, approach density in orange, and overlap shown by darker coloring (key in top-left corner). Histograms around the colormaps show the state density projected onto the longitudinal (top) and lateral (left) axes of the arena. Left: correct trials. Right: incorrect trials. (**B**) Investigation approach index (I.A.I.) for correct (purple) and incorrect (green) trials. Thick lines and shaded region are

*Figure 10 continued on next page*

Figure 10 continued

mean ± s.e.m., thin lines are individual mice. (C) Difference in I.A.I. between correct and incorrect trials along the longitudinal axis (2.5 cm bins, n = 9). Thick line is the across-mouse mean difference, thin gray lines are 1000 permutations in which correct and incorrect trial labels were scrambled. (D) Investigation occupancy along the lateral axis, within the transition zone (5–10 cm longitudinal) for correct and incorrect trials. Thick lines and shaded region are mean ± s.e.m., thin lines are individual mice. (E) Approach occupancy along the lateral axis, within the transition zone (5–10 cm longitudinal) for correct and incorrect trials. Thick lines and shaded region are mean ± s.e.m., thin lines are individual mice. (F) I.A.I. along the lateral axis, within the transition zone (5–10 cm longitudinal) for correct and incorrect trials. Thick lines and shaded region are mean ± s.e.m., thin lines are individual mice. (G) Difference in investigation occupancy between correct and incorrect trials along the lateral axis, within the transition zone. Thick blue line is the across-mouse mean difference, thin blue lines are 1000 permutations in which correct and incorrect trial labels were scrambled. (H) Difference in approach occupancy between correct and incorrect trials. Thick orange line is the across-mouse mean difference, thin orange lines are 1000 permutations in which correct and incorrect trial labels were scrambled. (I) Difference in I.A.I. between correct and incorrect trials. Thick orange line is the across-mouse mean difference, thin orange lines are 1000 permutations in which correct and incorrect trial labels were scrambled.

The online version of this article includes the following figure supplement(s) for figure 10:

**Figure supplement 1.** Occupancy maps indicate an advantage for investigation of both sides for both stay trials and switch trials.

Lockery, 2011). In contrast, other organisms, such as flying insects, often operate in a highly turbulent regime where concentration gradients are not reliably informative (Crimaldi et al., 2002; Murlis et al., 1992; Riffell et al., 2008). By design, mice in our task operate in an intermediate regime, where turbulent odor plumes close to the ground form noisy gradients (Gire et al., 2016; Riffell et al., 2008). By varying the absolute concentration and the concentration difference between the two sides, we tested whether performance in this regime is guided or gated by odor. Because behavior varies with the gradient and not the absolute concentration (Figure 2C–E), we have shown that mice are guided by gradient cues in this regime. Further, performance is higher when the mice sample both sides of the midline, suggesting that they sense the gradient by comparing sniff sequences across time.

Our naris occlusion experiments demonstrate that performance is statistically indistinguishable with naris occlusion, suggesting that stereo olfaction does not play a major role in our task. This finding contrasts with previous studies of olfactory navigation in a different regime: following a depositional odor trail. In these studies, stereo manipulations had small but significant effects on performance, and led to changes in movement strategy (Jones and Urban, 2018; Khan et al., 2012). Importantly, a study of olfactory search in moles showed that stereo reversal did not affect navigation at a distance from the target, but reversed turning behavior in the target's immediate vicinity (Catania, 2013). These results suggest that stereo cues may be informative near a source, where gradients are steep, but that stereo cues play less of a role at a greater distance from the source where gradients are more shallow. In this more distant condition, serial-sniff comparisons have been hypothesized as a potential sensory computation for odor gradient following Catania, 2013. We propose that our task design, in which mice must commit to a side at a distance from the source, forces mice out of the stereo regime and into the serial-sniff comparison regime. Neurons sensitive to sniff-to-sniff odor concentration changes have been observed in the olfactory bulb of head-fixed mice (Parabucki et al., 2019), providing a potential physiological mechanism for this sensory computation.

On the other hand, physiological mechanisms revealed in head-fixed mice may not generalize to the freely moving search condition. The external stimulus obtained by moving the nose through a noisy gradient differs dramatically from the square odor pulses delivered during head-fixed or odor-poke olfactory tasks. Further, the sniff statistics we observe in our mice are qualitatively faster than those reported in head-fixed mice under most conditions (Bolding and Franks, 2017; Shusterman et al., 2011; Wesson et al., 2009). One exception is that mice sniff fast in response to a novel odor (Wesson et al., 2009). Such fast stimulation impacts the responsiveness of olfactory sensory neurons (Esclassan et al., 2012; Ghatpande and Reisert, 2011; Verhagen et al., 2007). In addition to the temporal properties of odor transduction, short- and long-term synaptic and network plasticity mechanisms will influence the olfactory bulb's responses during fast sniffing (Beshel et al., 2007; Díaz-Quesada et al., 2018; Gupta et al., 2015; Jordan et al., 2018; Mandairon and Linster, 2009; Patterson et al., 2013; Zhou et al., 2020). Without tapping into the fast sniffing regime, the understanding we can gain from head-fixed studies in olfaction will be incomplete at best. In the

future, it will be necessary to complement well-controlled reductionist behavioral paradigms with less-controlled, more natural paradigms like ours.

Mice execute a strategic behavioral program when searching, synchronizing fast sniffing with three-dimensional head movements at a tens of milliseconds timescale. It has long been known that rodents investigate their environment with active sniffing and whisking behaviors (*Kepecs et al., 2006*; *Wachowiak, 2011*; *Welker, 1964*). More recent work has established that under some conditions sniffing locks with whisking, nose twitches, and head movement on a cycle-by-cycle basis (*Kurnikova et al., 2017*; *Moore et al., 2013*; *Ranade et al., 2013*). Sniffing also synchronizes with brain oscillations not only in olfactory regions, but also in hippocampus, amygdala, and neocortex (*Karalis and Sirota, 2018*; *Kay, 2005*; *Macrides et al., 1982*; *Vanderwolf, 1992*; *Yanovsky et al., 2014*; *Zelano et al., 2016*). Respiratory central pattern generators may coordinate sampling movements to synchronize sensory dynamics across modalities with internal brain rhythms (*Kleinfeld et al., 2014*). Further, locomotor and facial movement, which are often synchronized to respiration, drive activity in numerous brain regions, including primary sensory areas (*McGinley et al., 2015*; *Musall et al., 2019*; *Niell and Stryker, 2010*; *Stringer et al., 2019*). Why are respiration and other movements correlated with activity in seemingly unrelated sensory regions? In the real world, sensory receptors operate in closed loop with movement (*Ahissar and Assa, 2016*; *Gibson, 1966*). Consequently, sensory systems must disambiguate self-induced stimulus dynamics from changes in the environment. Further, active sampling movements can provide access to sensory information that is not otherwise available to a stationary observer (*Gibson, 1962*; *Schroeder et al., 2010*; *Yarbus, 1967*). Widespread movement-related signals may allow the brain to compensate for and capitalize on self-induced stimulus dynamics (*Poulet and Hedwig, 2006*; *Sommer and Wurtz, 2008*; *Sperry, 1950*; *von Holst and Mittelstaedt, 1950*; *Webb, 2004*). Our work advances understanding of how sensation and movement interact during active sensing.

Rigorously quantifying the behavior of freely moving animals is more feasible than ever, thanks to recent developments in machine vision, deep learning, and probabilistic generative modeling (*Datta et al., 2019*; *Gomez-Marin et al., 2014*; *Mathis and Mathis, 2020*), as our work shows. In particular, the motifs we have defined provide a compact description of the behavior, while still capturing the idiosyncrasies of individual mice. Importantly, these motifs can be grouped into two larger-scale behavioral states that we putatively call 'investigation' and 'approach'. Two-state search strategies are common across phylogeny (*Bargmann, 2006*; *Berg, 2000*; *Kennedy and Marsh, 1974*; *Lockery, 2011*; *van Breugel and Dickinson, 2014*; *Vickers and Baker, 1994*). In smaller organisms, state switches have provided a useful behavioral readout for understanding the neural mechanisms of odor-guided behavior (*Bi and Sourjik, 2018*; *Larsch et al., 2015*; *Baker et al., 2018*). Here, we have shown that where switches between investigation and approach occur in allocentric space can reveal the location of informative features in an olfactory scene. The transition points between 'investigation' and 'approach' serve as a principled template against which to compare neural activity. Our work thus establishes a framework for studying neural mechanisms of active sensing in an unrestrained mammal.

# Materials and methods

## Key resources table

| Reagent type (species) or resource | Designation | Source or reference | Identifiers | Additional information |
|---|---|---|---|---|
| Software, algorithm | Bonsai | Open Ephys<br>*Lopes et al., 2015* | | Visual reactive programming |
| Software, algorithm | Deeplabcut | The Mathis Lab of Adaptive Motor Control<br>*Nath et al., 2019* | | Animal pose estimation |
| Software, algorithm | Pyhsmm | Matthew Johnson,<br>*Johnson et al., 2013a* and *Johnson et al., 2013b* | | Bayesian inference in HSMMs and HMMs |

Custom-written task control, analysis, and visualization code is available at https://github.com/SmearLab/Freely-moving-olfactory-search (*Findley et al., 2021*).

## Animals: housing and care

All experimental procedures were approved by the Institutional Animal Care and Use Committee (IACUC) at the University of Oregon and are compliant with the National Institutes of Health Guide to the Care and Use of Laboratory Animals. C57BL/6J mice (2–14 months old) from the Terrestrial Animal Care Services (TeACS) at University of Oregon (19 males, 7 females) were used for behavioral experiments. Mice were housed individually in plastic cages with bedding and running wheels provided by TeACS. Mice were fed standard rodent chow ad libitum and were water-restricted, receiving a daily allotment (1–1.5 mL) of acidified or chlorinated water. Animal health was monitored daily, and mice were taken off water restriction if they met the 'sick animal' criteria of a custom IACUC-approved health assessment.

## Behavioral assay design

### Arena and task structure

Mice were trained to perform a two-choice behavioral task where they must locate an odor source for a water reward. This 15 × 25 cm behavioral arena was largely custom-designed in lab (all designs available upon request). The behavioral arena contains a custom-designed and 3D-printed honeycomb wall through which continuous clean air is delivered to the arena and a latticed wall opposite to the honeycomb allowing airflow to exit the arena. Two odor tubes (Cole-Parmer Instrument Company, #06605-27) are embedded inside the honeycomb wall and consistently deliver either clean or odorized air. There are three nose pokes in the arena: one trial initiation poke and two reward pokes. The initiation poke is embedded inside the latticed wall (where airflow exits) and is poked to initiate trials. The left and right reward pokes are embedded in the left and right arena walls against the honeycomb airflow delivery and are used for water reward delivery. Mice initiate odor release by entering the initiation poke. If the mouse locates the odor source successfully (by entering the quadrant of the arena containing the correct odor port), water (~6–8 μL) is available at the corresponding nose poke. An ITI of 4 s is then initiated. If the mouse goes to the incorrect side, water is not made available and they must wait an increased ITI of 10 s.

### Odor delivery

Odor is delivered to the arena using two custom-designed and built olfactometers. For a single olfactometer, air and nitrogen are run through separate mass flow controllers (MFCs) (Alicat Scientific, #MC-100SCCM-RD) that can deliver 1000 mL/min and 100 mL/min at full capacity, respectively. We can use these MFCs to control the percentage of total nitrogen flow (100 mL/min) that runs through liquid odorant. Consequently, we can approximately control the amount of odor molecules in the resulting odorized air stream. Total flow is maintained at 1000 mL/min (e.g., if we are delivering 80 mL/min of nitrogen, we will deliver 920 mL/min of air). Nitrogen MFC output is directed through a manifold (NResearch Incorporated, #225T082) with embedded solenoids that direct flow to one of four possible vials. These vials contain odorant diluted in mineral oil or are empty. To odorize air, nitrogen is directed through a vial containing liquid odorant. The nitrogen aerosolizes the odorant and combines with airflow MFC output at the exit point of the manifold. If nitrogen is directed through an empty vial, unodorized nitrogen will combine with airflow at the exit point. The resulting combined flow of air and nitrogen is then directed to a final valve (NResearch Incorporated, #SH360T042). Odorized air continuously runs to exhaust until this final valve is switched on at which point clean air is directed to exhaust and odorized air to the behavioral assay. Therefore, we can control the percentage of odorized flow (using the MFCs), the presence or absence of odorized flow (using the vials and solenoids), and the flow of odorized air to the assay (using the final valve). There are two olfactometers (one for each odor port), which are calibrated weekly to match outputs using a PID.

### Video-tracking

We use a Pointgrey Fly Capture Chameleon 3.0 camera (FLIR Integrated Imaging Solutions Inc, #CM3-U3-13Y3C) for video-tracking. We capture frames at 80 Hz at 1200 × 720 pixel resolution. All real-time tracking is executed using a custom Bonsai program. We isolate the mouse's centroid by gray-scaling a black mouse on a white background and finding the center of the largest object. We track head position by applying red paint on the mouse's implant between the ears and thresholding

the real-time HSV image to identify the center of the largest red shape. We can then identify nose position by calculating the extremes of the long axis of the mouse shape and isolating the extreme in closer proximity to the head point. These three points are sent to Python at 80 Hz for real-time tracking in our assay. We use this real-time tracking to determine successful odor localization; if the mouse enters the quadrant of the arena that contains the correct odor port, it has answered correctly. Bonsai is an open-source computer vision software available online (*Lopes et al., 2015*), and our custom code is available upon request.

For more rigorous behavioral analysis, we increased our tracking accuracy by using the open-source tracking software Deeplabcut (*Mathis et al., 2018*; *Mathis and Mathis, 2020*). All Deeplabcut tracking occurred offline following experimentation.

### Sniff recordings
We record sniffing using intranasally implanted thermistors (TE Sensor Solutions, #GAG22K7MCD419; see Materials and methods: Surgical Procedures). These thermistors are attached to pins (Assmann WSW Components, #A-MCK-80030) that can be connected to an overhead commutator (Adafruit, #736) and run through a custom-built amplifier (Texas Instruments, #TLV2460, amplifier circuit design available upon request).

### Software
All behavioral experiments were run using custom code in Python, Bonsai, and Arduino. Behavioral boards designed at Janelia Research Farms that use Arduino software and hardware were used to control all hardware. Bonsai was used to execute real-time tracking of animals, and Python was used to run the assay, communicate with Arduino and Bonsai, and save data during experiments. All programs used are open source, and all custom code is available upon request.

## Surgical procedures
For all surgical procedures, animals were anesthetized with 3% isoflurane; concentration of isoflurane was altered during surgery depending on response of the animal to anesthesia. Incision sites were numbed prior to incision with 20 mg/mL lidocaine.

### Thermistor implantation
To measure respiration during behavior, thermistors were implanted between the nasal bone and inner nasal epithelium of mice. Following an incision along the midline, a small hole was drilled through the nasal bone to expose the underlying epithelium ~2 mm lateral of the midline in the nasal bone. The glass bead of the thermistor was then partially inserted into the cavity between the nasal bone and the underlying epithelium. Correct implantation resulted in minimal damage to the nasal epithelium. The connector pins were fixed upright against an ~3 cm headbar (custom-designed and 3D printed) placed directly behind the animals' ears and the thermistor wire was fixed in place using cyanoacrylate. The headbar was secured against a small skull screw (Antrin Minature Specialties, #B002SG89OI) implanted above cerebellum. A second skull screw was placed at the juncture of the nasal bones to secure the anterior portion of the implant. All exposed skull and tissue were secured and sealed using cyanoacrylate. At the end of surgery, a small amount of fluorescent tempera red paint (Pro Art, #4435-2) was applied to the center of the headbar for tracking. Immediately following surgery, animals received 0.1 mg/kg buprenorphine followed by 3 days of 0.03 mg/kg ketoprofen. All but nine mice were implanted prior to training. Mice that were implanted post-training were taken off water restriction at least 2 days prior to surgery and were not placed back on water restriction for at least 1 week following all analgesic administration.

### Naris occlusion
To test the necessity of stereo olfaction as a sampling strategy, we occluded the nostrils of C57BL/6J mice using 6-0 gauge surgical suture (SurgiPro, #MSUSP5698GMDL). Mice were given 0.03 mg/kg ketoprofen and topical lidocaine on the nostril prior to induction. Suture was either pulled through the upper lip of the nostril and maxillary region to fully occlude the desired nostril or looped at the upper lip of the nostril for a sham stitch. Commercially available VetBond was applied to protect the suture knot. To ensure full occlusion, a small water droplet was placed on the occluded

nostril. The absence of bubbles or seepage indicated a successful occlusion. Occlusion was retested in the same manner directly before each experimental session. All stitches were removed within a week of application, and animals were stitched a total of three times per nostril.

## Behavioral training

All mice were trained to locate an odor source from one of two possible sources in the olfactory arena. Mice were removed from training and future experiments if they lost sniff signal or did not exceed 50 trials/perform above 60% correct in 15 sessions. The training process was divided into four primary stages.

### Water sampling

Mice were trained to alternate between the three pokes in the behavioral arena. Water (~5–8 µL) was made available at the nose pokes in the following order: initiation port, left reward port, initiation port, and right reward port (repeat). Mice were trained in this task for 30 min per session until the mouse completed 70 iterations. This took mice 2–9 sessions. Data are only shown for 19 mice because earlier iterations of the system did not save training data.

### Odor association

Mice were trained in the same sequence as water sampling. However, in odor association, water availability was removed from the initiation poke, and odor was released from whichever side water was available. Therefore, the mouse must initiate water availability by poking the initiation poke and then is further guided to the correct reward port by odor release. This task taught mice to initiate trials using the initiation poke and to associate odor with reward. However, in this step, odor is not required for reward acquisition as the task alternates left and right trials. Mice were trained in this task for 30 min per session until the mouse completed 70 iterations. This took mice 1–5 sessions. Data are only shown for 19 mice because earlier iterations of the system did not save training data.

### 100:0

Mice were given the same task as odor association, but with odor now randomly being released from the left or right odor port following an initiation poke. 10% of these trials were randomly 0:0 condition trials. To correctly answer, animals had to enter the quadrant of the arena (as tracked by the overhead camera) where odor was being released. If they answered correctly, water was made available at the reward port on the corresponding side. If they answered incorrectly, water was not made available and the mouse received an increased ITI. Mice were trained in this task for 40 min per session until they exceeded 80% accuracy, which took 1–4 sessions ($n = 26$).

### 80:20

When trials were initiated in this task, odor was released from both odor ports, but at differing concentrations. The animal had to enter the quadrant containing the odor port releasing the higher concentration. In this case, 80 means that the nitrogen MFC was set to 80 mL/min on one olfactometer (see Materials and methods: behavioral assay). Therefore, one odor port would release roughly 80% of the total possible odorant concentration. If one olfactometer was set to 80, then the other olfactometer would be set to 20 in this condition. 10% of these trials were randomly 0:0 condition. Mice were trained in this task for 40 min per session, taking 1–9 sessions to exceed 60% performance ($n = 24$).

## Behavioral experiments

### Variable ΔC, Constant |C|

This experiment tested how performance and sampling strategy changes with task difficulty. In this experiment, mice performed a two-choice behavioral task where they located an odor source for a water reward at varying concentration differences between the two ports. This experiment interleaved several possible conditions: 100:0 (all odor released from one port or the other), 80:20 (odor is released from both ports at different concentrations: 80% of the total possible airborne concentration and 20% of the total possible airborne concentration), and 60:40 (60% and 40%). Additionally, there was a control condition where all system settings were the same as the 80:20 condition, but

nitrogen flow was directed through a clean vial so that the final flow was not odorized. 10% of the total number of trials were the 0:0 condition. Mice ran 40 min experimental sessions and totaled 5–50 sessions ($n = 19$). These experiments were run with 1% liquid dilution of pinene.

### Novel odorant

This experiment tested how mice generalized our olfactory search task. A subset of mice were run with 1% liquid dilution of vanillin, which, unlike pinene, does not activate the trigeminal system ($n = 3$).

### Constant ΔC, Variable |C|

This experiment tested if the animals use a thresholding strategy based on a fixed concentration threshold to solve the localization task. We ran this experiment using air dilution delivering the concentration ratios 90:30 and 30:10 interleaved randomly ($n = 5$). Mice ran 40 min sessions, and we analyzed data from the first session.

### Naris occlusion

This experiment tested the necessity of stereo olfaction in our localization task. Mice were run in the interleaved experiment (see above) initially. However, after observing no differences between concentration groups, we continued this experiment running mice in the 80:20 and 0:0 conditions only. Mice were run in one of five categories: left occlusion, left sham stitch, right occlusion, right sham stitch, and no stitch (see Materials and methods: surgical procedures). Mice ran 40 min experimental sessions and totaled 5–30 sessions ($n = 13$). Stitches were always removed after 4 days. These experiments were run using 1% pinene dilutions.

## Mapping the olfactory environment

We used a PID (Aurora Scientific Inc, #201A) to capture real-time odor concentration at a grid of 7 × 5 sampling locations in the assay. Using vials of 50% liquid dilution of pinene, we captured ~15 two-second trials per sampling point. Odor maps were generated using the average concentration detected across all trials at each location. These maps were smoothed via interpolation across space. Discriminability maps in *Figure 1—figure supplement 2C, D* were calculated with ROC analysis on the PID data (*Green and Swets, 1966*). To generate the distributions, each 2 s trial was divided into 25 ms chunks (approximately the mean inhalation duration during the task). For each space bin, the mean value of each 25 ms chunk was compiled into a distribution of odor concentration values for right and left trials (the different gradient conditions were pooled for this analysis). To map concentration gradient discriminability, 25 ms samples from each bin were assembled into a pseudosample, such that each sampling position had a concentration value. The gradient angle in each bin of this pseudosample was then calculated (imgradient function in MATLAB) and compiled into a distribution of angles for right and left trials (the different gradient conditions were pooled for this analysis). For both absolute concentration and gradient maps, the area under the ROC curve was calculated for each bin, scaled to between −1 and 1, and absolute valued, and these were assembled into a map and smoothed. Values are thresholded and shown at low bit depth (eight grayscale values) to facilitate perception of where the auROC values are highest.

## Data analysis

Analyses of odormaps, sniffing, DLC tracking, and motif sequences were performed in MATLAB. Inhalation and exhalation times were extracted by finding peaks and troughs in the temperature signal after smoothing with a 25 ms moving window. Sniffs with duration less than the 5th percentile and greater than the 95th percentile were excluded from analysis. For alignment of movement with sniffing, tracking and motif sequences were shifted forward in time by 25 ms (two frames), the temporal offset revealed by video calibration (*Figure 1*).

### *Figure 1*

Odormaps were visualized by smoothing the PID sampling grid with a Gaussian and colored using Cubehelix (*Green, 2011*).

## Figure 2

Sessions where mice performed less than 60% correct on 80:20 (90:30 for Constant ΔC, Variable |C|) were less than 80 trials or had any missing folders or files were excluded. Trials longer than 10 s were excluded. Percent correct was calculated by dividing the correct trials by total trials in a single session and was averaged across all sessions, all mice. Trial duration was measured between nose poke initiation and reward poke and was averaged across all trials, all sessions, all mice. Tortuosity was measured by dividing the total path length by the shortest possible path length and was averaged across all trials, all sessions, all mice.

Statistical tests were performed in Python using the scipy package (*Peterson et al., 2001*). A binomial test was used to test statistical significance of above-chance performance. Wilcoxon rank-sum tests were used for all group comparisons with pairwise comparisons for more than two groups. Two group comparisons were tested using all trials pooled together, and pairwise comparisons of three groups or more were tested across mice using individual mouse averages.

## Figure 3

Occupancy and sniff rate colormaps were generated by down-sampling the tracking data to a 50 × 30 grid of bins (0.5 cm²). Occupancy colormaps are a 2D histogram of the nose position data. Sniff rate histograms were generated by dividing the sniff count in each position bin by the corresponding bin in the occupancy histogram. Both histograms were Gaussian-smoothed and colored using Cubehelix (*Green, 2011*). Grand means are shown in *Figure 3F, G*, while individual mouse occupancy heatmaps are shown in *Figure 4*. Maps were colored using Cubehelix (*Green, 2011*).

## Figures 4 and 5

Nose speed, yaw velocity, and Z-velocity were calculated from the three-point position time series generated by Deeplabcut. For analysis, a 400 ms window centered on each inhalation time was extracted from the kinematic time series. Colormaps in *Figure 4* show traces surrounding individual sniffs, while colormaps were generated using Bluewhitered (*Childress, 2020*). For within-trial sniffs, only those inhaled before the decision line were included. The ITI sniffs are taken from the time of reward port entry to the time of the first initiation port entry in the ITI. For cross-correlation and coherence analysis, we aligned the time series of sniffing and kinematic parameters from the entire trial or from the interval between reward and initiation port in the ITI. Tracking glitches were excluded by discarding trials or ITIs that contained frames with nose speed above a criterion value (100 pixels per frame).

## Figure 6

Average motif shapes were generated from the mean positions of the nose, head, and body points from the first eight frames of every instance of a given motif as determined by the AR-HMM. Decoding analysis is described in the following section.

## Figure 7

The transition probability matrix was clustered by minimizing Euclidean distance between rows. For analyses separating investigation and approach sniffs, sniffs were defined as investigation or approach sniffs based on the state at the inhalation time. Colors for investigation and approach were selected from the Josef Albers painting, Tautonym, (B) (*Albers and Tautonym, 1944*).

## Figure 8

Figures are generated by motif-onset triggered averages of inhalation times determined as described above. *Figure 8B, C* are the grand mean of the motif onset-triggered average for each motif. Maps were colored using Cubehelix (*Green, 2011*). Sniff phase (relative time in sniff) was determined by dividing the motif onset latency from inhalation by the total duration (i.e., inhalation time to inhalation time) of each sniff. Modulation index was calculated as the difference between maximum and minimum instantaneous sniff rate, divided by the sum ($max - min/max + min$).

*Figure 9*, *Figure 9—figure supplement 1*, *Figure 10*, *Figure 10—figure supplement 1*

Investigation and approach occupancy maps were generated by down-sampling the tracking data to a 25 × 15 grid of bins (1 cm$^2$). Occupancy maps are a 2D histogram of the nose position data, compiled separately for investigation and approach frames (see below for details of ARHMM analysis). In plots where the data are reoriented with respect to the choice, the lateral axis of all right-choice trials has been flipped so that the trajectories always end on the left side (top side in the displayed occupancy maps). Both histograms were normalized to the total occupancy in a given bin (i.e., investigation + approach), Gaussian-smoothed, and merged and colored using a scheme adapted from fluorescence microscopy (*Geissbuehler and Lasser, 2013*). Grand means (*n* = 9) are shown in *Figures 9A, B* and *10A*, and S16, while individual mouse mean occupancy maps are shown in *Figure 9—figure supplement 1*. I.A.I. is calculated as the difference between investigation and approach occupancies over their sum for a given bin. I.A.I. is taken from histograms that are the projection of the 2D maps onto the longitudinal or lateral axes. The 'transition zone' is defined as the region between 5 and 10 cm from the longitudinal axis origin (i.e., the initiation port), and lateral axis histograms are taken from within this region in *Figure 10D–F* and *Figure 10—figure supplement 1*. Correct-incorrect occupancy and index differences are grand mean of the individual mouse differences in *Figure 10C, G–I* and *Figure 10—figure supplement 1*. These differences are evaluated statistically against a null distribution generated by scrambling the correct and incorrect trial labels 1000 times and re-running this analysis. Importantly, these shuffles are performed within mice before taking the post-shuffle grand means, so that these null distributions incorporate both within-mouse and across-mouse variability.

## Sniff synchronization

Sniff cycles were compared with kinematics to determine the extent of movement modulation at individual sampling points. Individual sniffs were cross-correlated with each kinematic signal (i.e., nose speed) at −200 ms from inhalation onset to +200 ms from inhalation onset. To further determine synchrony between the two signals, we measured the coherence of signal oscillation between sniff signals and individual kinematic measurements at −200 ms from inhalation onset to +200 ms from inhalation onset.

## Auto-regressive hidden Markov model

Let $x_t$ denote the six-dimensional vector of nose-head-body coordinates at video frame $t$ (sampled at 80 Hz), with components $(x_{\text{nose}}, y_{\text{nose}}, x_{\text{head}}, y_{\text{head}}, x_{\text{body}}, y_{\text{body}})$. We fit an AR-HMM to mouse trajectory data, $\left\{\mathbf{x}_t^{(i)}\right\}$, across trials (indexed by *i*) from 13 out of 15 mice (two mice were excluded a priori due to low task performance). These mice performed olfactory search under the following experimental conditions: Variable ΔC, Constant |C| (nine mice); naris occlusion (seven mice); and Constant ΔC, Variable |C| experiments (five mice).

### The generative view

Viewed as a generative model (that generates simulated data), the AR-HMM has two 'layers': a layer of hidden discrete states (corresponding to discrete movement motifs) and an observed layer that is the continuous trajectory $x_t$. We denote the temporal sequence of discrete states by $z_t$. In each time step, $z_t \in \{1, 2, \ldots, S\}$, that is, it is one of an *S* number of states, or movement motifs. The discrete hidden states evolve in time according to a Markov chain: going from time step *t* to *t* + 1, the discrete state may change to another state according to a transition probability matrix $\pi_{z_1, z_2}$, which denotes the conditional probability of switching to $z_2$ having started in $z_1$. The probability distribution over the initial state, $z_{t=1}$, of the Markov chain at the start of each trial was taken to be the uniform distribution.

Now suppose for time steps $t_1$ to $t_2$ (inclusive) the discrete layer remained in state *z*. The continuous or auto-regressive (AR) part of the model dictates that, over this time interval, the continuous trajectory, $x_t$, evolves according to a linear AR process. The parameters of this AR process can be different in different states or motifs, *z*. In other words, $x_t$ is governed by

$$\boldsymbol{x}_t = A_z \boldsymbol{x}_{t-1} + \boldsymbol{b}_z + \varepsilon_t. \quad t_1 \leq t \leq t_2$$

where $A_z$ is a $6 \times 6$ matrix and $b$ is a $6 \times 1$ vector, and the noise vector $\varepsilon_t$ is sampled from the multivariate zero-mean Gaussian distribution $N(0, Q_z)$, where $Q_z$ is a $6 \times 6$ noise covariance matrix. Moreover, the parameters $A_z$, $\boldsymbol{b}_z$, and $Q_z$ depend on the discrete state $z$, and in general are different in different discrete states. The simple stochastic linear dynamics described by *Equation (1)* can describe simple motions of the mouse, such as turning left/right, dashing towards a certain direction, freezing (when $A_z$ is the identity matrix and $\boldsymbol{b}_z$ is zero), etc. The switches between these simple behaviors allow the model to generate complex trajectories.

The AR-HMM is an example of a model with latent variables, which in this case are the discrete state sequence $z_t^{(i)}$ in each trial. The model, as a whole, is specified by the set of parameters $(\pi, \{A_z\}, \{\boldsymbol{b}_z\}, \{Q_z\})$, which we will denote by $\theta$. For a $d$-dimensional trajectory ($d = 6$ here) and $S$ states, comprises $S(S-1) + S(d^2 + d + d(d+1)/2) = S(S-1 + 3d(d+1)/2)$ parameters.

## Model fits

Models with latent variables are often fit using the expectation-maximization (EM) algorithm, which maximizes the likelihood of the model in terms of the parameters $\theta \equiv (\pi, \{A_z\}, \{\boldsymbol{b}_z\}, \{Q_z\})$ for a given set of observed data $\left\{\boldsymbol{x}_t^{(i)}\right\}$. In this work, we did not use the EM algorithm, but adopted a fully Bayesian approach in which both the hidden variables and the model parameters were inferred by drawing samples from their posterior distribution (*Wiltschko et al., 2015*). The posterior distribution combines the model likelihood and Bayesian priors imposed on its parameters, according to Bayes' rule. If we denote the joint likelihood of observed trajectories, $\left\{\boldsymbol{x}_t^{(i)}\right\}$, and the latent variables, $\left\{z_t^{(i)}\right\}$, by $P\left(\left\{\mathbf{x}_t^{(i)}, z_t^{(i)}\right\}|\theta\right)$ and the prior distribution over model parameters by $P(\theta)$, then up to normalization, the joint posterior distribution of latent variables and model parameters is given by

$$P\left(\left\{z_t^{(i)}\right\}, \theta \Big| \left\{\boldsymbol{x}_t^{(i)}\right\}\right) \propto P\left(\left\{\left(\boldsymbol{x}_t^{(i)}, z_t^{(i)}\right)\right\}|\theta\right) P(\theta).$$

For the AR-HMM, the (logarithm of the) joint log-likelihood is given by

$$\log P\left(\left\{\boldsymbol{x}_t^{(i)}, z_t^{(i)}\right\}|\theta\right) = \sum_i \sum_{t=2}^{T_i} \left[\log \pi_{z_{t-1}^{(i)}, z_t^{(i)}} + log N\left(\boldsymbol{x}_t^{(i)}|A_{z_t^{(i)}}\boldsymbol{x}_{t-1}^{(i)} + \boldsymbol{b}_{z_t^{(i)}}, Q_{z_t^{(i)}}\right)\right].$$

where $T_i$ is the length of trial $i$, and we use the notation $N(\boldsymbol{x}|\mu, Q) = e^{-\frac{1}{2}(x-\mu)^T Q^{-1}(x-\mu)} / \sqrt{|2\pi Q|}$ to denote the density at point $x$ of a multivariate Gaussian with mean vector μ and covariance matrix $Q$.

We imposed loose conjugate priors on the model parameters, which were factorized over the parameters of the AR process, $(\{A_z\}, \{\boldsymbol{b}_z\}, \{Q_z\})$, in different discrete states $z$, and the different rows of the Markov transition matrix, $\pi$. On the rows of $\pi$, we imposed Dirichlet distribution priors with uniform distribution means and concentration hyperparameter $\alpha$, which was set to 4. We imposed matrix normal inverse Wishart priors on the AR parameters, independently for different discrete states. Under this prior, the noise covariance $Q_z$ has an inverse Wishart distribution with a 'scale matrix' hyperparameter, which was set to the $d \times d$ ( = $6 \times 6$) identity matrix, and a 'degrees-of-freedom' scalar hyperparameter set to $d + 2 = 8$. Conditional on $Q_z$, the remaining AR parameters, $(A_z, \boldsymbol{b}_z)$, have a joint multivariate normal distribution under the prior, which can be specified by the prior mean and joint prior covariance matrix of $A_z$ and $b_z$. The prior means of $A_z$ and $\boldsymbol{b}_z$ were set to the $d \times d$ identity matrix and the $d$-dimensional zero vector, respectively, while the prior covariance matrix of the concatenation $(A_z, \boldsymbol{b}_z)$ was given by the tensor product of $Q_z$ and the $(d + 1) \times (d + 1)$ ( = $7 \times 7$) identity matrix (equivalently, under this prior, $b_z$ and different columns of $A_z$ are independent and uncorrelated, while each of these column vectors has a prior covariance equal to the [prior] AR noise covariance, $Q_z$).

Bayesian model inference was carried out by sampling from (instead of maximizing) the joint posterior distribution of the model parameters and latent state variables conditioned on the observed trajectory data (*Equation 2*). We did this by Gibbs sampling (an example of Markov

chain Monte Carlo; not to be confused with the Markov chain in the AR-HMM), which works in a manner conceptually similar to the EM algorithm: it switches between sampling $z_t^{(i)}$ in all trials, conditioned on previously sampled parameters, and then sampling the parameters $\theta$ given the previous sample of $\left\{z_t^{(i)}\right\}$. To carry out this model inference procedure, we used the Python package developed by M.J. Johnson and colleagues, publicly available at https://github.com/mattjj/pyhsmm (*Johnson et al., 2013a*).

We ran the Gibbs sampler for 300 iterations and burned the first 200 samples, retaining 100. We used the remaining samples to obtain the posterior probabilities of hidden discrete states at each time step of each trial (by calculating the frequency of different state in that time step and trial, across the retained Gibbs samples), as well as posterior expectation of the model parameters (by calculating their averages over the retained Gibbs samples). We refer to the AR-HMM with parameters given by these latter posterior expectations as the 'fit model'.

## Model selection

We fit AR-HMM's with different numbers of states (motifs), $S$, to mouse trajectory data pooled across animals. To evaluate the statistical goodness of fit of these fit model and select the best $S$ (the number of states or motifs), we evaluated the log-likelihoods of fit models on trajectory data from a held-out set of trials, not used for model fitting. The corresponding plot of log-likelihoods is shown in *Figure 6F*. As seen, the log-likelihood keeps increasing with $S$, up to $S = 100$. This shows that, up to at least $S = 100$, additional motifs do have utility in capturing more variability in mouse trajectories. These variabilities may include differences in movement across mice, as well as movement variations in the same mouse but across different trials or different instances of the same movement; for example, a clockwise head turn executed with different speeds in different instances or trials. In the AR-HMM, the AR observation distribution of a given Markov state corresponds to a very simple (linear) dynamical system that cannot capture many natural and continuous variations in movement, such as changes in movement speed. Nevertheless, AR-HMM models with higher $S$ can capture such variations with more precision by specializing different discrete Markov states, with different AR distributions, to movement motifs of different mice, or, for example, to capture different speeds of the same qualitative movement motif.

The goal for this modeling was to give a compact description of recurring movement features across animals and conditions, suitable for visualization and alignment. For these purposes, the goodness of fit did not provide a suitable criterion because the log-likelihood plots did not peak or plateau even at very large numbers of states. Guided by visual inspection, we thus chose the model with $S = 16$ for the main figures (*Figures 6–8*). Although this was a somewhat arbitrary choice, we show that the findings in *Figures 6–8* do not depend on the choice of $S$ – models with $S = 6$, 10, or 20 gave equivalent results (*Figure 6—figure supplements 2–4*).

## MAP sequences

The Gibbs sampling algorithm that we used for model inference yields (time-wise marginal) maximum a posteriori (MAP) estimates of the latent variables $\left\{z_t^{(i)}\right\}$, as follows. Using the Gibbs samples for the latent variables, we can estimate the posterior probability of the mouse being in any of the $S$ states in any given time step of a given trial. We made MAP sequences by picking, at any time step and trial, the state with the highest posterior probability. The inferred MAP motifs tended to have high posterior probability, which exceeded 0.8 in 66.2% of all time steps across the 17,195 trials in the modeled dataset.

## Decoding analysis

We decoded experimental conditions and animal identities from single-trial MAP motif sequences inferred using the AR-HMM. Specifically, we trained multi-class decoders with linear decision boundaries (linear discriminant analysis) to decode the above categorical variables from the single-trial empirical state transition probability matrices derived from the MAP sequence of each trial. If $z_t^{\hat{(i)}}$ is the motif MAP sequence for trial $i$, the empirical transition probability, $\pi_{a,b}^{\hat{(i)}}$, from state $a$ to state $b (a, b \in \{1, \ldots, K\})$, for that trial was calculated by

$$\hat{\pi}_{a,b}^{(i)} \equiv \frac{n_{a,b}^{(i)}}{\sum_{c=1}^{K} n_{a,c}^{(i)}}.$$

$$n_{a,b}^{(i)} \equiv \sum_{t=1}^{T^{(i)}-1} I(\hat{z}_t = a) I(\hat{z_{t+1}} = b).$$

where $T^{(i)}$ is the length of trial $i$, and $I(\cdot)$ is an indicator function, returning 1 or 0 when its argument is true or false, respectively.

We used the decoder to either classify experimental condition or mouse identity, in different trials (*Figure 6D, E*). For decoders trained to classify the trials' experimental condition, we used pooled data across mice. For decoders trained to classify mouse identity, we only used data from the 80:20 odor condition. Data was split into training and test dataset in a stratified fivefold cross-validation manner, ensuring equal proportions of trials of different types in both datasets. The trial type was the combination of left vs. right decision, experimental condition, and mouse identity.

To calculate the statistical significance of decoding accuracies, we performed an iterative shuffle procedure on each fold of the cross-validation. In each shuffle, the training labels that the classifier was trained to decode were shuffled randomly across trials of the training set, and the classifier's accuracy was evaluated on the unshuffled test dataset. This shuffle was performed 100 times to create a shuffle distribution of decoding accuracies for each fold of the cross-validation. From these distributions, we calculated the z-score of decoding accuracy for each class in each cross-validation fold. These z-scores were then averaged across the folds of cross-validation and used to calculate the overall p-value of the decoding accuracy obtained on the original data.

## Acknowledgements

We thank Z Mainen for initiating this area of research with MS and for advising on interpretation of behavioral results, B Datta for advising on AR-HMM analysis, L Mazzucato for advice on the decoding analysis, S Shoham for advising on allocentric analyses, and P Gupta for suggesting behavioral experiments. We thank A Singh Bala, C Niell, S Lockery, M Wehr, and B Datta for helpful comments on the manuscript. MS was supported by grants from the NIH (R56DC015584, R21NS104935, and R34NS116731), the Whitehall Foundation, and start-up funds from the University of Oregon. TF was supported by an NIH fellowship (F31DC016799). MB was supported by an NIH fellowship (F32MH118724). DW and YA were supported by start-up funds from the University of Oregon.

## Additional information

### Funding

| Funder | Grant reference number | Author |
|---|---|---|
| Whitehall Foundation | 2015-12-201 | Matthew C Smear |
| National Institutes of Health | R56DC015584 | Matthew C Smear |
| National Institute of Neurological Disorders and Stroke | R21NS104935 | Matthew C Smear |
| National Institute of Neurological Disorders and Stroke | R34NS116731 | Matthew C Smear |
| National Institute on Deafness and Other Communication Disorders | F31DC016799 | Teresa M Findley |
| National Institute of Neurological Disorders and Stroke | F32MH118724 | Morgan A Brown |
| University of Oregon | Start up funds | Morgan A Brown Matthew C Smear |

The funders had no role in study design, data collection and interpretation, or the decision to submit the work for publication.

### Author contributions
Teresa M Findley, Yashar Ahmadian, Conceptualization, Data curation, Software, Formal analysis, Supervision, Investigation, Visualization, Methodology, Writing - original draft, Project administration, Writing - review and editing; David G Wyrick, Conceptualization, Data curation, Software, Formal analysis, Validation, Investigation, Visualization, Methodology, Writing - original draft, Writing - review and editing; Jennifer L Cramer, Investigation, Methodology; Morgan A Brown, Resources, Software, Investigation, Methodology, Writing - review and editing; Blake Holcomb, Investigation, Methodology, Writing - original draft; Robin Attey, Software, Formal analysis, Investigation, Methodology; Dorian Yeh, Eric Monasevitch, Nelly Nouboussi, Isabelle Cullen, Investigation; Jeremea O Songco, Jared F King, Supervision, Investigation; Matthew C Smear, Conceptualization, Resources, Data curation, Software, Formal analysis, Supervision, Funding acquisition, Validation, Investigation, Visualization, Methodology, Writing - original draft, Project administration, Writing - review and editing

### Author ORCIDs
Teresa M Findley ⓘ https://orcid.org/0000-0002-2050-4869
David G Wyrick ⓘ https://orcid.org/0000-0001-8096-5766
Robin Attey ⓘ http://orcid.org/0000-0002-9652-8103
Yashar Ahmadian ⓘ https://orcid.org/0000-0002-5942-0697
Matthew C Smear ⓘ https://orcid.org/0000-0003-4689-388X

### Ethics
Animal experimentation: This study was performed in strict accordance with the recommendations in the Guide for the Care and Use of Laboratory Animals of the National Institutes of Health. All of the animals were handled according to approved institutional animal care and use committee (IACUC) protocols (AUP-17-23) of the University of Oregon. All surgery was performed under sodium isofluorane anesthesia, and every effort was made to minimize suffering.

### Decision letter and Author response
Decision letter https://doi.org/10.7554/eLife.58523.sa1
Author response https://doi.org/10.7554/eLife.58523.sa2

## Additional files

### Supplementary files
• Transparent reporting form

### Data availability
Source code is available on github at https://github.com/Smear-Lab/Olfactory_Search (copy archived at https://archive.softwareheritage.org/swh:1:rev:fcb2c2aa1a4438f22d622f29b01a0c64e8e4df85), and source data files are uploaded to Dryad.

The following dataset was generated:

| Author(s) | Year | Dataset title | Dataset URL | Database and Identifier |
|---|---|---|---|---|
| Smear M, Findley T, Wyrick D, Ahmadian Y | 2021 | Sniff-synchronized, gradient-guided olfactory search by freely-moving mice | http://dx.doi.org/10.5061/dryad.r7sqv9sc0 | Dryad Digital Repository, 10.5061/dryad.r7sqv9sc0 |

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
