## [Decision Letter]

**Acceptance summary:**

This paper is a clear account of an odor-guided behavior in which the authors use machine-learning movement analysis to work out how mice combine odor sampling with a set of sniff-locked movement motifs in their decision-making. The authors find that in this task, the mice use odor gradients, but do not use stereo olfaction. The careful characterization of movement motifs during the task will be useful to relate olfactory decision-making with neuronal activity.

**Decision letter after peer review:**

Thank you for submitting your article "Sniff-synchronized, gradient-guided olfactory search by freely-moving mice" for consideration by *eLife*. Your article has been reviewed by 3 peer reviewers, one of whom is a member of our Board of Reviewing Editors, and the evaluation has been overseen by Catherine Dulac as the Senior Editor. The reviewers have opted to remain anonymous.

The reviewers have discussed the reviews with one another and the Reviewing Editor has drafted this decision to help you prepare a revised submission.

As the editors have judged that your manuscript is of interest, but as described below that additional analysis is required before it is published, we would like to draw your attention to changes in our revision policy that we have made in response to COVID-19 (https://elifesciences.org/articles/57162). First, because many researchers have temporarily lost access to the labs, we will give authors as much time as they need to submit revised manuscripts. We are also offering, if you choose, to post the manuscript to bioRxiv (if it is not already there) along with this decision letter and a formal designation that the manuscript is "in revision at *eLife*". Please let us know if you would like to pursue this option. (If your work is more suitable for medRxiv, you will need to post the preprint yourself, as the mechanisms for us to do so are still in development.)

Summary:

This paper is a clear account of an odor-guided behavior in which the authors use machine-learning movement analysis to characterize the behavior in detail. The key findings are sniff-synchronized movement (already known), the ability to classify a number of movement motifs (but not strikingly distinct) and the further analysis of relationships between these movements and sniffing.

All reviewers felt that this detailed analysis of behavior in a non-invasive manner was exciting and has much promise for the field.

Essential revisions:

The reviewers felt that the interpretation of sampling and movement needed a better understanding of the strategy used by the mice.

They suggest that there are several possibilities:

– The animals could be memorizing absolute concentrations.

– The animals could use two samples during the turn, to ascertain gradients.

Further, they felt that the motif analysis might be useful to elucidate how the animal corrects an initial decision.

They feel that the authors need to provide the reader with a detailed and rigorous analysis of the decision strategy. As the discussion on this topic was extensive, I have provided excerpts below to help the authors.

In addition the authors need to do more with the analysis, clarifying:

– Animal differences and lack of stereotypy in movements;

– Nose speed during ITI;

- State transitions and error correction;

- Decision points.

Here I provide excerpts from the extensive discussions on this paper. The intention is to give the authors an understanding of the key points that the reviewers took from the paper, and where they felt that its analysis could be strengthened.

After watching the videos several times, here is my interpretation of the decision strategy: when the trial starts, the mouse faces away from the gradient. It must make a rotation of ~180 degrees to align its body with the direction of the gradient. This stereotyped movement forms an arc of the alpha shape of Figure 3D. While rotating, the mouse could already detect whether the concentration increases or not. Upon the detection of an increase, the mouse infers that the gradient points in that direction, and it initiates a walk toward the corresponding odor port. During that walk/approach, the mouse sniffs a couple of times. If a decrease in concentration is measured during these sniffs/samples, the mouse might still be able to stop and reorient before the decision boundary.

In addition, I don't think that a sequence of samples on the left and the right side must be taken for the mouse to infer the direction of the gradient. All it needs to do is compare the intensity when the head is aligned with the body (before ) with the intensity after a lateral sample (after). If an increase is detected, the head must be pointing toward the gradient. In the videos, I didn't see systematic left-right samples.

…

sampling a landscape with two longitudinal asymmetric gradients would be challenging. This situation would make the sensory experience associated with an alpha turn largely inconclusive. Without making multiple samples on the left and right sides of the midline, a mouse would not be able to obtain a coarse map of the landscape to inform its decision.

But the intensity landscapes reported in Figure S2 indicate that most gradients do not really have two real maxima (or two lobes). Even for the 60:40 condition, the landscape essentially looks like one smooth gradient with a maximum on one side. So the sensory experiences produced by an alpha turn toward the left and the right side might be different – a signal the mouse might learn?

That said, I was puzzled by the fact that the two landscapes corresponding to the 60:40 conditions are quite different when 60 is located on the left or the right side (top versus bottom panels in Figure S2B). So, one might expect that the 60:40 – right would be harder to scan than 60:40 – left. Since the performances of the left and the right conditions were lumped together, it's impossible to tell whether this prediction is correct.

And here is another possible strategy:

An alternative possibility is that animals are not actually making L-R comparisons and just memorizing 'expected' gradients or absolute concentrations (100, 80, 60 etc) – which is actually quite easy for mice to learn!

In that case, indeed the mouse actually already knows which side the reward will be. If during the alpha turn it smells concentrations 100 or 80 or 60, it sticks to that side, otherwise walk to the other side. Indeed in this scenario – there is no need for lateral comparisons – and the knowledge gathered during the alpha turn already tells the animals which side it should go towards.

But this in principle is a different task than the authors intended to set up!

The authors actually do try to rule out the possibility that animals learn absolute concentrations by doing what they refer to as variable |C| sessions (Figure S4 B). i.e. by present the same absolute concentration, but in opposing contexts – one where its the higher of the two concentrations (30:10) versus one where its the lower of the two concentrations (30:90).

But the data presented is not really conclusive –.…

performance in the first 10 trials is quite low – so its very likely that the animals just learn a new rule..

*Reviewer #1:*

This paper is a clear account of an odor-guided behavior in which the authors use machine-learning movement analysis to characterize the behavior in detail. The key findings are sniff-synchronized movement (already known), the ability to classify a number of movement motifs (but not strikingly distinct) and the further analysis of relationships between these movements and sniffing.

The basic behavioural checks and controls are thoroughly done. It is interesting that there is no stereo component to these decisions.

A key finding is sniff-synchronized movement. This has also been seen in other studies (Kurnikova et al. 2017, Moore et al. 2013, Ranade et al. 2013) as the authors point out. I was looking for a clear statement of how the current work advances this understanding.

Motif analysis.

The motifs don't appear to be particularly crisp, in that they continue to contribute up to 100 motifs. I was looking to see this enumerated, as in % of variance explained (or in this case cross-validated log-likelihood). It turns out it is done in Figure S8C. This should be in the main text.

There is nice but not much explored finding of there being distinct movement patterns between individual mice.

The motif correlation to stage of trial and to sniffing and nose speed is interesting but maybe not surprising. The subsequent analysis shows up a number of patterns here which are suggestive of general synchronization between breathing and other motor rhythms. I wonder if the authors could do a zero-order correlation, of something simple like leg movements which are much more directly quantifiable than these motifs. Or has such work been done?

The main accomplishment, to my reading, is the detailed characterization of sniffing and its relation to movement. The authors are candid about the being mostly a descriptive account of behavior and movement and make a case for this being a prerequisite for subsequent mechanistic and interventional studies.

On the one hand, I appreciate the value of a thorough descriptive account of freely moving behavior. However, it seems to me that the motifs are fuzzy and the core outcome of sniff-locked movement has been reported. I wonder if there is more to be gleaned from this rich dataset, such as an analysis of what differs between mice or whether there is something underlying the lack of stereotypy in the movements of the mice.

*Reviewer #2:*

In this study, Smear et al. aim to investigate how mice sample the noisy stimulus information from olfactory plumes such as to navigate towards their source. To this end, they developed a 2AFC task for freely moving mice where the same odor emanates from two lateral sources, at independently controllable concentrations. Mice are required to identify the more intense of the two sources and collect water from reward port located on the side with the higher odor concentration. The authors improve on previous attempts at studying this problem by requiring the mice to commit to their decision at a substantial distance from the odor ports. This forces the mice to assess odor concentration from distal cues rather than via serial sampling of the sources themselves. Interestingly, the authors find that stereo olfaction (comparing concentration across two nostrils) is not required to determine source location from distal cues. Using a series of stimulus conditions, the authors convincingly show that in their paradigm, mice rely on olfactory cues and specifically the relative, not absolute, concentration difference between the two sources.

The relevance of stereo olfaction for airborne odor cues has been long debated. In my opinion, the authors results in principle resolve this debate – stereo comparisons allow finer source localisation near the source, while serial sampling may play a larger role farther away from source. One concern however is that this lack of reliance on stereo sampling may result from the specific task design and the constraints it imposes on the behavior (see concerns).

Further, the authors characterise the sampling behaviors of mice during this task by monitoring respiration (thermistor) as well as nose, head and body positions (video tracking). The authors find a striking, active synchronisation of sniffing and nose (and body) movements that gets selectively recruited during putative investigatory phase of the task i.e. when mice are actively exploring the concentration gradient. The authors do exhaustive analysis to show that such synchronisation is not a default state and the coupling is much weaker during other phases within the same trial. While such coupling of movement and intrinsic rhythms has been proposed previously, to my knowledge this is the first careful characterisation of this phenomenon in freely moving mice. Importantly, the authors results not only confirm the existence of such coupling but also clarify that this synchronisation is an active feature of olfactory navigation. Interestingly however, the authors do not find any significant difference in sampling strategies across different stimulus difficulties (see concerns).

Lastly, the authors use machine learning to parse motion trajectories into identifiable behavioral motifs. With their approach, they find that a range of motifs that are stereotyped across mice and occur in non-random sequences during each trial. Further, a trained decoder can successfully decode mouse identity from the sequences in which these motifs in each animal. While these motif based analysis are well done, the data presented do not seem to make any clear predictions about how these motif sequences would change in different task conditions. The authors do not find any obvious relationship between trial types (difficult versus easy stimuli) and motif sequences and the presented analyses do not add much to the main message of the paper. I therefore lack the imagination to accurately assess the relevance of this portion of the study.

Overall, the study is well executed – the data presented are clear with numerous controls at each step. In my opinion, the evidence provided for lack of need for stereo olfaction for distal source assessment and active synchronisation of sniffing and sampling movements are important contributions to the field of olfaction that warrant publication in *eLife*. However, I have several conceptual concern about the task design and the interpretations of the results that the authors should clarify prior to publication.

1. My primary concern is about the task design. I commend the authors for the careful control of olfactory stimuli and substantial improvements over previously published odor localisation assays by separating the reward port from the odor source and forcing decisions at locations distant from the source. However, the task design chosen does not really require the mice to localize odor sources beyond just indicating whether there is more odor on the right versus left. This is different from natural conditions, where the necessary spatial resolution may be much higher. In fact, finer source localization confers no additional benefit for maximizing reward in this task. Therefore, the sampling strategies exhibited by the subjects here may be different from those employed during natural odor navigation where the motivation is to precisely locate the source of mate, food or predators.

2. Along the same lines, it is surprising that even for the easiest version of the task, the performance hovers around 80%, even though PID characterisations show very clear differences between the two halves of the arena. Furthermore, performance drops with increasing stimulus difficulty but mice do not appear to change their sampling strategies to compensate for the lower reward rate. I am trying to reconcile these two facts. At first pass, given known olfactory acuity of rodents, it seems that trained mice should have no trouble handling the easiest stimuli (reach almost 100% success). One possibility is that the mice are not fully motivated/engaged in the task. An alternate explanation is that mice are fully motivated to reach maximum reward rate, but the task is just too hard and the sampling strategies employed at 100:0 condition are their best, and therefore with increasing stimulus difficulty, they cannot perform any better. Yet, the latter possibility appears very unlikely. Can the authors comment on these two possibilities? The question remains whether mice when pushed to achieve higher performances would employ different sampling strategies.

3. Lastly, looking at Figure 5Ai it appears that overall nose speeds are significantly lower during ITI than during the investigatory bouts within the trial. While the authors rule out that sniffing-movement synchronisation is simply a feature of rapid sniffing, they do not rule out the dependence on running speeds. Perhaps the apparent lack of synchrony in ITI results from poorer ability to resolve decelerations, given lower speeds on average. This is also consistent with reduced, but significant synchrony during premature initiations in the ITI (Figure S7A) where speeds tend to be higher than those shown in Figure 5Ai. This should be easily addressable by repeating the analysis on speed matched datasets.

*Reviewer #3:*

In this manuscript, Findley and colleagues propose a novel assay to study the behavioral strategy freely moving mice adopt to navigate turbulent odor gradients. This assay is neat and well-thought. It sheds light into the control of active sampling through sniffing and search patterns involving head and body movements. The work is an admirable technical tour de force. The results are built on solid data analysis, which makes use of unsurprised machine learning to avoid subjective biases in the categorization of behavioral states. The manuscript offers a wealth of data that should be a wide interest to the field of olfaction. Finally, the conclusions are presented in a way that is balanced and supported by the well-controlled experimental data. This manuscript combines innovation with rigor to advance our understanding of olfaction in rodents (and beyond).

I have a few suggestions to improve the manuscript. These suggestions do not require any additional experiments.

1. An exciting finding of the manuscript is the description of two behavioral states underlying odor search: investigation and approach. The authors might want to push the analysis of the search strategy one step further by defining whether/how mice can switch from investigation to approach, back to investigation to perform error correction. This process would rule out that animals find the gradient through an initial guess that leads to a full commitment to one side during the approach phase. The data suggests that error correction takes place (Figure 7C and D), but those cases are not analyzed in detail. Can a statistical analysis of the state transitions reveal any principles in the organization of error correction? Does the animal's state indeed switch from approach to investigation during error correction?

2. The occupancy diagram of Figure 3F is fascinating. Together with panel 3D, it suggests that mice undergo fairly stereotyped searches: after poking their nose out of the initiation port, they appear to make a 180 degree rotation (sweep) to face the gradient. The density reaches a maximum at that point (opposite to the position of the initiation port). Is this position (crossing of the alpha shape) dominated by an "investigation" state? Can this position be viewed as a decision point? When/where does the animal tend switch to the "approach" state? More generally, could you map dominate trends in behavioral motifs of Figure 6B onto the stereotyped alpha shape of the occupancy diagram of Figure 3F?

3. Figure 2D: Could you speculate about the reason why trials tend to be longer for the 100:0 conditions compared to the more difficult 60:40? Do mice spend more time in the investigation phase when gradient is stronger? Although this result would be counter-intuitive, it might suggest that mice learn (?) to spend less time on the initial search when less information is available to them.

4. Figure 1D indicates that the average gradient's geometry is not the same when the odor is delivered on the left side (100:0) compared to the right side (0:100). This observation appears to be true for the other odor ratios reported in Figure S2. This asymmetry should affect the gradient that the animal experiences during the investigation phase, which should in turn influence the accuracy of the decisions. Do you expect a 50:50 condition to produce no preference (on average)?

5. Figure 7F: What are you concluding from this panel? Are you sure that the shaded areas represent the standard deviation and not the SEM? If the standard deviation is shown, how do you explain the existence of stereotypical wiggles on a timescale of 50 ms? It would be very useful to represent a variant of Figure 7C where the trials are sorted between correct and incorrect.

---

## [Author Response]

SummaryThis paper is a clear account of an odor-guided behavior in which the authors use machine-learning movement analysis to characterize the behavior in detail. The key findings are sniff-synchronized movement (already known), the ability to classify a number of movement motifs (but not strikingly distinct) and the further analysis of relationships between these movements and sniffing. All reviewers felt that this detailed analysis of behavior in a non-invasive manner was exciting and has much promise for the field.

We are gratified by the overall positivity of the review and we are most grateful to the reviewers for thoughtful and thought-provoking comments. We recognize and appreciate the effort and time these reviews must have taken. Most importantly, we feel that we have substantially improved our manuscript by responding to these reviews.

We have organized our response into sections that address the major issues raised in Dr. Bhalla’s summary. To do so, we have organized the comments from individual reviewers into these sections. All comments about other issues are addressed after that.

Essential revisionsDecision points and sensory strategy. Where and when are the mice making decisions, and are these decisions based on absolute concentration or the gradient?

We thank the reviewers for encouraging us to delve deeper into the sensory strategy the mice are using to solve this task. In our revision, we now present data that support our contention that the mice are guided by serial sniffs across odor gradients. To further test this model, we visualized the allocentric structure of investigation and approach, as suggested by reviewer 3.

The occupancy diagram of Figure 3F is fascinating. Together with panel 3D, it suggests that mice undergo fairly stereotyped searches: after poking their nose out of the initiation port, they appear to make a 180 degree rotation (sweep) to face the gradient. The density reaches a maximum at that point (opposite to the position of the initiation port). Is this position (crossing of the alpha shape) dominated by an "investigation" state? Can this position be viewed as a decision point? When/where does the animal tend switch to the "approach" state? More generally, could you map dominate trends in behavioral motifs of Figure 6B onto the stereotyped alpha shape of the occupancy diagram of Figure 3F?

We thank reviewer 3 for these ideas. In a new section of the results with 4 new figures (2 main and 2 supplementary), we now show occupancy maps of investigation and approach states. By overlaying the occupancy histograms from the two states, we show that most of the overlap is restricted to the center of the region between initiation port and decision line. This is at the crossing of the alpha shape of occupancy maps (Figure 9A) where overall occupancy also peaks (Figure 3F). To better quantify this overlap, we show an index of the relative values of investigation and approach occupancy as a function of distance from the initiation port. These data show that the index switches from predominantly investigation to predominantly approach within the alpha shape crossing region, between 5 and 10 cm from initiation. Based on this observation, we now refer to this region (between 5 and 10 cm), as a "transition zone" for the purposes of further analysis. We do not view this region as a "decision point", because we think that term only applies to an instantaneous event, not an across-trial pattern. Investigation-approach transitions are our best guess at the decision points in each individual trial. What we are calling the "transition zone" is the region where most of these transitions occur.

We next use these occupancy maps to evaluate alternative models of the sensory strategy – do the mice use absolute concentration or gradients? As argued by Dr. Bhalla:

The animals could be memorizing absolute concentrations… just memorizing 'expected' gradients or absolute concentrations (100, 80, 60 etc) – which is actually quite easy for mice to learn! In that case, indeed the mouse actually already knows which side the reward will be. If during the alpha turn it smells concentrations 100 or 80 or 60, it sticks to that side, otherwise walk to the other side. Indeed in this scenario – there is no need for lateral comparisons – and the knowledge gathered during the alpha turn already tells the animals which side it should go towards.

We agree that this possibility is logically consistent with the evidence provided in the original submission. We thank Dr. Bhalla for articulating such a clear prediction of the absolute concentration model. Indeed, an absolute concentration-sensing mouse would not need lateral comparisons. If they first turn toward higher concentration, they should stick to that side, and proceed straight to the water port, rather than waste time and sniffs at the midline. They would only need to cross the midline if they sense low concentration while turning out of the initiation port. To depict this intuitive prediction, we have now added auROC maps based on our PID recordings as panel C and D in Figure 1—figure supplement 2. This map (Figure 1—figure supplement 2C) shows absolute concentration differences between left and right trials can best be discriminated if the animal samples directly downwind of the odor ports, along the axes of maximal odor concentration.

On the other hand, if the mouse is using a gradient sensing strategy, it would seem that the best strategy is to sample both sides. As stated by Dr. Bhalla:

I don't think that a sequence of samples on the left and the right side must be taken for the mouse to infer the direction of the gradient… In the videos, I didn't see systematic left-right samples.

We agree that sampling both sides is not necessary for performing the task. However, because the airflow in the arena is turbulent, odor released on one side spreads some distance into the other. Nevertheless, while sampling both sides is not a must, we agree that the gradient sensing model predicts that the animal would achieve the highest performance by sampling across the midline. To depict this intuition, in (Figure 1—figure supplement 2D) we show an auROC map based on gradients derived from the same PID recordings. This map shows that for gradient-sensing, the most informative place to sample is indeed across the midline.

In Figure 10, we use investigation and approach occupancy maps to test these predictions. We show that correct trials are associated with more investigation at and around the midline, particularly on the unchosen side of the arena. By definition, sampling the unchosen side precedes crossing the midline to the chosen side, showing that correct trials feature more sampling on both sides. Incorrect trials have more investigation downwind of the odor ports, particularly near the decision line, where Figure 1—figure supplement 2D shows is the most informative location for absolute concentration discrimination. Thus, these data are inconsistent with the absolute concentration model, and consistent with the gradient model.

One concern with our task is that it forces mice to turn in one direction or the other out of the initiation port. Sometimes the mouse stays on the side it started towards, sometimes it switches to the other side. Could this asymmetry explain why correct trials seem to feature more investigation at the midline? In Figure 10—figure supplement 1, we analyzed stay and switch trials separately. The performance correlations shown in Figure 10 are essentially the same. In both conditions, correct trials are associated with more investigating at the midline and on the unchosen side. Although there are intriguing differences between the patterns, we feel that these are beyond the scope of the present manuscript.

Our analysis of the allocentric structure of state usage supports a serial-sniff gradient sensing model and is inconsistent with an absolute concentration sensing model. On that same topic:

The authors actually do try to rule out the possibility that animals learn absolute concentrations by doing what they refer to as variable |C| sessions… But the data presented is not really conclusive – performance in the first 10 trials is quite low – so its very likely that the animals just learn a new rule.”

We respectfully disagree. Figure 2—figure supplement 2 shows the across-mouse average performance over trials. In the first 10 trials of the first session in which the mice have encountered the 90:30/30:10 version of the task, the mice perform at 75% correct. Therefore, on average, the mice make only 2 or 3 errors in these trials. Even for an ideal observer mouse, we think it would take at least two errors to ascertain that they should learn a new rule. Slotnick and Katz (1974) showed that rats can show learning-set performance for new odor-pair discriminations in a nearly-errorless way, but only after they have already experienced 16 previous odor-pair switches, and thousands of trials. So, while it may be mathematically possible for a mouse to immediately change sensory strategies from C to delta-C, the possibility does not seem very likely given our data.

State transitions and error correction

The authors might want to push the analysis of the search strategy one step further by defining whether/how mice can switch from investigation to approach, back to investigation to perform error correction. This process would rule out that animals find the gradient through an initial guess that leads to a full commitment to one side during the approach phase. The data suggests that error correction takes place (Figure 7C and D), but those cases are not analyzed in detail. Can a statistical analysis of the state transitions reveal any principles in the organization of error correction? Does the animal's state indeed switch from approach to investigation during error correction?

Our new analysis shows that transitions from investigation to approach in the region just before the decision line are more common on error trials (Figure 10C and supplement 1). This pattern shows that the investigation state is not inherently beneficial to performance. Instead, it matters where the mouse investigates, which gives us indication of where informative stimulus features are in the arena. The auROC map (Figure 1—figure supplement 2) shows that absolute concentration features are most informative in this position near the decision line, where investigation is associated with incorrect trials. Thus we interpret this result as further evidence against the absolute concentration model.

What are individual animal differences and how do you explain the lack of stereotypy in movement?

We have shown that a classifier can uniquely identify individual animals based on our ARHMM. Is this because the motifs themselves differ across mice, or does it reflect diversity in how different mice sequence and deploy the motifs? To test the former possibility, in Figure 6—figure supplement 5 we present average shapes of each motif (as in Figure 6B) for each individual mouse. The shapes match across mice, suggesting that the algorithm is identifying consistent behavioral features. Instead, the differences have more to do with where and when the mice deploy different motifs. To accompany Figures 9 and 10, we provide investigation and approach occupancy maps for individual mice in Figure 9—figure supplement 1, showing that mice are diverse in where they transition from investigation and approach. For example, some mice are biased to one side, other mice to the other. We think these are the idiosyncrasies that the classifier is picking up on. Interestingly, we show in Figure 9—figure supplement 1 that if trials are re-oriented with respect to the chosen side (i.e., right-choice trials are flipped so that the trajectory always ends on the upward side of the diagram), all the mice tend to transition from investigation to approach on the chosen side. In this sense, the mice are quite consistent.

We think it is most likely that the lack of stereotypy in individual trial trajectories is attributable to the variable and turbulent nature of our odor stimuli.

Additional comments

1. Poor performance and lack of adaptive strategy:

We too were surprised that the mice did not perform better in the 100:0 condition. Additionally, Reviewer #2 points out mice do not adapt their strategy as the task is made presumably more difficult (i.e. from 80:20 to 60:40). We can only speculate as to why.

First, we don’t know that this is necessarily an easy task for rodents. In previous studies by the Bhalla and Murthy groups, the rodents were able to improve performance by using a memory-guided strategy, and to some degree avoid the problem of tracking the odor source from a distance. Maybe the animals in these studies used memory-guided strategies because odor-guided navigation is harder for them than our intuition and our PID maps would suggest. We know frustratingly little about the statistics of natural olfactory scenes, so perhaps the mice are evolutionarily optimized to operate in different stimulus conditions than we have contrived for this paradigm.

Alternatively, another likely explanation which we have added to the manuscript (lines 460-464) is that mice are strong delay-discounters – they are in a hurry to collect as much reward as possible in as little time as possible. Perhaps if we could more exhaustively search the task parameter space (e.g., ITI durations, reward sizes), we could find a way to slow them down and improve performance, but this has exceeded our experimental bandwidth so far.

2. Nose speed during ITI versus trial:

Reviewer #2 raised the concern that our analysis of sniff-locked movement with regards to nose speed is biased, because of differing nose speeds during the inter-trial interval and the trial. In Figure 5—figure supplement 2, we now separately analyze ITI sniffs in which the mice were moving at or above the average nose speed during the trials. Even in these sniffs, we see very little modulation of nose speed, and none for yaw or z-velocity, consistent with our assertion that sniff-synchronized movement is a pro-active search strategy, and not a default accompaniment of fast locomotion.

3. Asymmetrical odor distribution:

Reviewers and Dr. Bhalla pointed out that there is asymmetry in the right versus left odor delivery, particularly apparent in our 60:40 PID maps. This asymmetry is due to a difference in airflow distribution across the arena. If this asymmetry were relevant to performance, we would see systematic patterns in the position biases across mice. However, our data show no such patterns in the left-right distribution of occupancy as far as we can tell (see Figures 3-supplement 1, 9-supplement 1).

4. Sniff-synchronization novelty:

We have been asked to provide a statement on how our sniff-synchronization finding expands upon what is already known, particularly from the work of Kleinfeld’s group. Most importantly, our task design, which includes trial and ITI periods, allows us to show that kinematic rhythms do not always lock to the sniff cycle. Instead, we show that sniff-synchronized movement is specific to periods when the mouse is searching. From this we can infer that sniff synchronization is a pro-active sampling behavior rather than an odor-gated orientation reflex or a default accompaniment to fast sniffing. This could only be speculated upon in the previous work.

5. Ethological gap between task design and natural conditions:

Reviewer #2 points out that our binary choice-based task design may not require the fine spatial resolution likely needed for olfactory search in natural conditions where the number of possible target locations is quite a bit larger than 2. We agree and acknowledge this limitation of our study. We also have no doubt that there are many features of olfactory search that we cannot capture with a paradigm like this. Despite these limitations, we feel confident that the primary findings of the study – gradient guidance, sniff synchronization, and two-state organization of search behavior – will hold true under more naturalistic conditions of airborne scent tracking. We hope we and others can improve upon this experimental design to better recapitulate the relevant olfactory features and motor affordances of the real world.